# Diffusionless transformation of soft cubic superstructure from amorphous to simple cubic and body-centered cubic phases

Jie Liu[1,2], Wenzhe Liu [3], Bo Guan[4], Bo Wang [3], Lei Shi [3], Feng Jin [1], Zhigang Zheng[5✉], Jingxia Wang [1,2✉], Tomiki Ikeda[1] & Lei Jiang [1,2]

In a narrow temperature window in going from the isotropic to highly chiral orders, cholesteric liquid crystals exhibit so-called blue phases, consisting of different morphologies of long, space-filling double twisted cylinders. Those of cubic spatial symmetry have attracted considerable attention in recent years as templates for soft photonic materials. The latter often requires the creation of monodomains of predefined orientation and size, but their engineering is complicated by a lack of comprehensive understanding of how blue phases nucleate and transform into each other at a submicrometer length scale. In this work, we accomplish this by intercepting nucleation processes at intermediate stages with fast cross-linking of a stabilizing polymer matrix. We reveal using transmission electron microscopy, synchrotron small-angle X-ray diffraction, and angle-resolved microspectroscopy that the grid of double-twisted cylinders undergoes highly coordinated, diffusionless transformations. In light of our findings, the implementation of several applications is discussed, such as temperature-switchable QR codes, micro-area lasing, and fabrication of blue phase liquid crystals with large domain sizes.

[1] CAS Key Laboratory of Bio-inspired Materials and Interfaces Sciences, Technical Institute of Physics and Chemistry, Chinese Academy of Sciences, Beijing, China. [2] Center of Material Science and Optoelectronics Engineering, School of Future Technology, University of Chinese Academy of Sciences, Beijing, China. [3] Department of Physics, Key Laboratory of Micro-and Nano-Photonic Structures, and State Key Laboratory of Surface Physics, Fudan University, Shanghai, China. [4] Institute of Chemistry, Chinese Academy of Sciences, Beijing, China. [5] Department of Physics, East China University of Science and Technology, Shanghai, China. ✉email: zgzheng@ecust.edu.cn; jingxiawang@mail.ipc.ac.cn

Soft cubic phases have distinct crystal symmetries and interconnected three-dimensional periodic structures[1–7], providing opportunities for the development of photonic crystals[8,9], next-generation optical devices[10], and energy technologies[11–13]. Blue-phase liquid crystals (BPLCs) are typical soft cubic phases[8,14–19] that combine the order of solid molecular crystals and the fluidity of the liquids. BPLCs have attracted intense scientific and engineering interest in recent decades because they exhibit narrow photonic bandgaps and sub-millisecond response times, do not require alignment layers, and have potential applications in adjustable lasers[20–26], displays[27–29], and nonlinear optical devices[30,31]. BPLCs are analogous to atomic crystals[14,17,32,33] based on their highly ordered structures at both the molecular (orientational molecular order) and mesoscopic scales (chiral arrays of double-twisted cylinders (DTCs), which are induced by the self-assembly of the molecules). The DTCs stack spontaneously to fill a three-dimensional space, forming a simple cubic (SC) lattice (blue phase II, BPII) with a space group of $O^2$ ($P4_232$)[8,10,21,34–36] or a body-centered cubic (BCC) lattice (blue phase I, BPI) with a space group of $O^8$ ($I4_132$)[8,10,34,35,37,38]. BPLCs are obtained by slow cooling from the isotropic state, and a phase transformation occurs from blue phase III (BPIII) to BPII or BPI, in which BPIII is thought to consist of a spaghetti-like tangle of DTCs[15,39,40]. The characteristic lattices of BPI and BPII are several hundred nanometers in size[36,38,41] allowing for light manipulation at the visible, infrared, or ultraviolet wavelengths. Various optical devices[17,42–49] have been fabricated based on the phase-transition process of BPLCs. To date, the phase transformation of BPLCs has been characterized using polarized optical microscopy (POM)[17,42–49]. The DTCs, which are analogous to the atoms of the atomic crystals, can be considered as structural units of BPIII[15,39,40], BPII[8,10,31,34–36], or BPI[8,10,34,35,37,38], in which their microstructures have been observed using confocal laser scanning microscopy[41,50], transmission electron microscopy (TEM)[38,51], and corresponding simulations[32,52,53]. However, the phase-transition process has yet to be observed at the sub-micrometer scale owing to the poor stability of the transition states, and thus the transition mechanism remains unclear. A full understanding of the phase-transition processes of the soft cubic superstructure is expected because it is highly important not only for fundamental science but also for practical applications, and will provide important insights into the design and fabrication of functional materials and devices.

In this work, TEM, synchrotron small-angle X-ray diffraction (syn-SAXS), and angle-resolved microspectroscopy (ARM) were used to dynamically track the phase transformation process of BPLCs: including diffusionless phase transformations (DLPTs) of BPIII-to-BPII, BPIII-to-BPI, and reversible thermoelastic martensitic of BPII-to-BPI. In this case, the DTCs are considered as structural units that do not diffuse during the DLPTs. The intermediate stages with core-shell configurations are fast polymer-stabilized to achieve ultra-high thermal stability from −190 to 340 °C for further characterization. Besides, the successive DLPTs and dual-stage formation mechanism of a thermoelastic martensitic transformation are confirmed.

## Results and discussion

**Diffusionless phase transitions from polarized optical microscopy.** BPLCs with the following chemical compositions were fabricated: HTG135200/C6M/R5011/trimethylolpropane triacrylate (TMPTA)/Irgacure 651 (I-651) = 30/61/3.5/4/1.5 (wt%) (the abbreviations and chemical structures of the chemicals are shown in Supplementary Tables 1 and 2). To fabricate the polydomain BPLCs, the mixtures were injected into a liquid crystal (LC) cell, followed by heating to 90 °C (~10 °C above the clearing point)

and stirred for 1 h. The BPLCs exhibit typical temperature windows of distinct phases: isotropic (>79.4 °C), BPIII (79.4 °C to 74.9 °C), BPII (78.0 °C to 74.9 °C), BPI (77.7 °C to 73.9 °C), and cholesteric phase (N*, <73.9 °C) (Supplementary Figs. 1–2). It is observed that the temperature windows of BPIII, BPII, and BPI overlap with each other, resulting in the coexistence of different phases and the formation of the core-shell configurations. The BPLCs then exhibit phase transitions through five stages (Fig. 1a_2–e_2 and Supplementary Figs. 1 and 3–4) as it is cooled at 0.05 °C/min. Furthermore, the coexisting textures in each stage are polymer-stabilized through photopolymerization for further characterization (Fig. 1a_3–e_3). Stage I shows BPIII embedded in an isotropic background at 79.4 °C forming crystal nuclei without any observable reflection signals (Fig. 1a). In this case, BPIII called blue fog, with an amorphous structure and dark blue color, is similar to that of the isotropic state[53] (Fig. 1a_2). The dark blue color gradually extends when the temperature falls from 79.4 °C (Supplementary Movie 1). In Stage II (Fig. 1b), bright blue domains (BPII) start to nucleate at the center of the dark blue domains (BPIII) at 78.0 °C, and the BPII core with a rounded square shape (Supplementary Fig. 5) grows anisotropically to form a BPIII/BPII core-shell configurations (Fig. 1b_2b_3 and Supplementary Figs. 5–6 and 7b–e). The appearance of the configurations are accompanied by an appearing reflection signal (Fig. 1b_4) with a central wavelength ($\lambda_c$) of 483 (463) nm before (after) polymerization. Clear interfaces between BPIII and BPII can be observed, and the crystal nucleus of BPII increases rapidly to full size within only a few seconds (Supplementary Figs. S5–6 and S8f_1–f_4) with the full size limited by the domains of BPIII. It was found that a thin shell of BPIII always exists between the isotropic state and BPII, and no BPII nuclei can directly appear in the isotropic state (Fig. 1d_2 and Supplementary Figs. 6f and 8f, g), suggesting that BPII cannot transfer directly from the isotropic state without pre-formed BPIII. Thus, it is predicted that the formation of BPII benefits from the DTCs in the mesophase BPIII (a spaghetti-like tangle of DTCs), undergoing a diffusionless reconfiguration of DTCs from BPIII. Herein, DTCs serve as building blocks that are analogous to atoms for atomic crystals[14,17,32,33]. The transition between BPII and BPI in Stage III (Fig. 1c) consists of hybrid phases of BPIII, BPII, and BPI. In this case, the BPLCs undergo DTC reconfiguration with crystal-line symmetry transformations of BPIII-to-BPI (Supplementary Fig. 8e–g), BPIII-to-BPII (Supplementary Figs. 5–7, 8a-b, f_1–f_4), and BPII-to-BPI (Supplementary Figs. 8c, d, g, h, 9–10), forming BPIII/BPII, BPIII/BPI, or BPIII/BPII/BPI (Supplementary Figs. 4 and 7f) core-shell configurations. The size (large or small) and random orientation of BPII and BPI domains are determined by the heterogeneous or homogeneous nucleation of BPIII inside the LC cell. It is found that green BPI domains with the {110} crystal plane parallel to the substrates (BPI_{110}) can directly nucleate within and grow from the center of the dark blue BPIII domains, forming BPIII/BPI core-shell configurations (Fig. 1c_2 and Supplementary Figs. 4, 8f, 9–10). As the temperature decreases further, parts of the residue BPIII belonging to the BPIII/BPI core-shell configurations transfer to BPII, forming BPIII/BPII/BPI core-shell configurations. During this process, an unusual phase transformation from BPIII to BPI appears earlier than that from BPIII to BPII (Supplementary Fig. 8f_1–f_4), which is probably caused by the tendency to preferentially form thermodynamically stable BPI. Similar to BPII, no BPI is observed to directly nucleate in the isotropic state, which may be attributed to the formation of BPI undergoing a diffusionless reconfiguration of the pre-formed DTCs in BPIII. Thus, the phase transformations of BPIII/BPI and BPIII/BPII are considered to be DLPT. At Stage III, two strong reflection peaks originating from the BPIII/BPII_{100}/BPI_{110} core-shell configurations appear spontaneously (Fig. 1c_4). In the

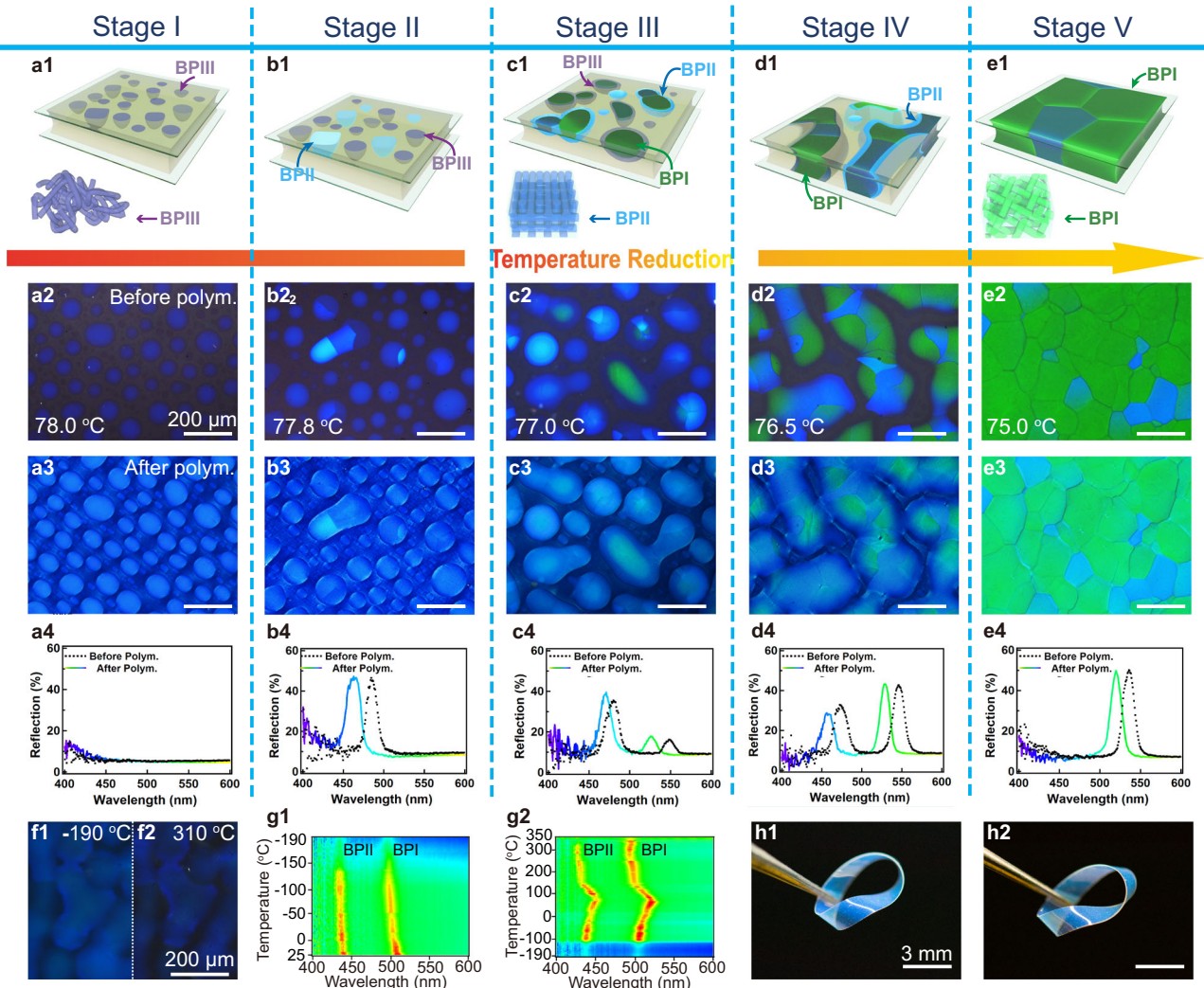

**Fig. 1 Evolution of the phase-transition process of BPLCs and the temperature stability of polymer-stabilized BPLCs (PS-BPLCs) in Stage IV. $a_1$–$e_1$** Schematic illustration of the phase-transition processes of **$a_1$** BPIII in Stage I; **$b_1$** BPIII domain and BPIII/BPII core-shell configuration, which coexist in Stage II, **$c_1$** BPIII domains and core-shell configurations of BPIII/BPII, BPIII/BPI, and BPIII/BPII/BPI coexist in Stage III; **$d_1$** BPIII/BPII/BPI core-shell configurations that exist in Stage IV; and **$e_1$** BPI domains in Stage V. **$a_2$–$e_2$, $a_3$–$e_3$** POM images of typical textures in distinct stages before **$a_2$–$e_2$** and after **$a_3$–$e_3$** polymerization. **$a_4$–$e_4$** Reflection spectra of samples during the phase change before and after polymerization. **$f_1$, $f_2$** Transmission-mode POM images of textures at −190 °C and 340 °C, respectively. **$g_1$, $g_2$** Temperature-dependent reflection spectra during the **$g_1$** cooling and **$g_2$** heating process. **$h_1$, $h_2$** Thermal stability of Mobius-strip-like PS-BPLCs in Stage IV. Optical photographs of the original strip **$h_1$** and the strip after ten cycles of immersion in liquid nitrogen **$h_2$**. "Before polym." and "after polym." refers to before and after polymerization. The phase-transition processes are divided into five stages according to the coexistence of distinct phases of BPLCs. Each stage is polymer-stabilized for further characterization. The PS-BPLCs show excellent thermal stability in a temperature window of −190 °C to 340 °C.

BPIII/BPII/BPI core-shell configurations, BPI nucleates in BPII, and the composite structure is then covered by BPIII shells (Fig. 1$c_2$–$c_3$). The photonic stopband, which corresponds to the green region with $\lambda_c = 550$ (526) nm before (after) polymerization, originates from BPI$_{\{110\}}$[25]. A peak in the blue domain with $\lambda_c = 477$ (462) nm before (after) polymerization corresponds to BPII with the {100} planes parallel to the substrate (BPII$_{\{100\}}$)[18]. In addition, the phase transformation of BPII/BPI is thought to be a martensitic transformation, which is the result of lattice-invariant shears through twinning (as anticipated for true martensite) (Supplementary Fig. 12a, b). Surface relief, as a result of the martensitic transformation, is also observed in both poly- and single-domain BPI during a phase transformation (Supplementary Fig. 12$c_1$–$c_3$). In addition, the phase transformation between BPI and BPII is reversible with little hysteresis (Supplementary Figs. 10–11 and Movie. 2). Stage IV (Fig. 1d) is obtained at 76.5 °C

and consists of hybrid phases of BPIII, BPII, and BPI simultaneously. Only the BPIII/BPII/BPI core-shell configurations and BPIII domains remain in Stage IV. The BPIII/BPI and BPIII/BPII core-shell configurations disappear, transferring to their BPIII/BPII/BPI counterparts. No more crystal nucleus of BPII or BPI is formed in the BPIII domains. Only the existing BPI and BPII nuclei formed in the previous stages in the BPIII/BPII/BPI core-shell configurations can grow. Here, two spectra appear in the BPIII/BPII/BPI core-shell configurations with $\lambda_c = 469$ (455) nm for BPII$_{\{100\}}$ or 546 (525) nm for BPI$_{\{110\}}$ before (after) polymerization (Fig. 1$d_4$). It was observed that the numbers of nuclei of both BPI and BPII are gradually reduced (Supplementary Fig. 13a–h) with an increasing annealing time for 24 h at Stage IV, accompanied by an enlargement of the domain sizes of the BPIII/BPII/BPI core-shell configurations. Finally, BPI domains with large sizes of over 1 mm (Supplementary Fig. 14d–e) were

obtained after the slow cooling of the enlarged BPIII/BPII/BPI core-shell configurations at 0.05 °C/min, in which the size is nearly 3-times larger than that of the sample without annealing (Supplementary Figs. 14a, b). Stage V (Fig. 1e) exhibits only a monophasic polydomain BPI with a large domain size (Fig. 1e$_2$-e$_3$) at 74.5 °C, and all BPII and BPIII were completely transferred to BPI, accompanied by a single reflection signal with $\lambda_c = 535$ (519) nm before (after) polymerization (Fig. 1e$_4$).

To comprehensively characterize the rearrangement of DTCs at the submicrometer scale, PS-BPLCs were obtained through the photopolymerization of BPLCs at different stages (Fig. 1a$_3$–e$_3$). It is worth mentioning that the DLPT properties are only observed in BPLCs that are not photopolymerized (Supplementary Fig. 16a$_1$–a$_5$). Once the BPLCs are photo-polymerized, the phase transformation cannot occur in the PS-BPLCs (Supplementary Fig. 16b$_1$–b$_5$). The as-prepared PS-BPLCs with a wide temperature window preserve the optical properties and microstructures of pristine BPLCs to directly track the rearrangement of DTCs at the microscale in a real- or reciprocal-space. For example, the samples of the PS-BPLCs obtained at Stage IV have BPIII/BPII/BPI core-shell configurations with a broad temperature window (Fig. 1f and Supplementary Fig. 17) from −190 to 340 °C. The broad temperature window of the PS-BPLCs was identified by cooling the sample from 25 to −190 °C and then heating it from −190 to 350 °C at 5 °C/min. The textures (from POM observations, Fig. 1f, and Supplementary Fig. 17) and reflection signals (Fig. 1g and Supplementary Fig. 17) of the sample change subtly during this large temperature change, and two reflection signals always coexist. Furthermore, the shape and color of the Mobius strip made of PS-BPLCs in Stage IV changed slightly after the strip was immersed in liquid nitrogen for ten times (Fig. 1h and Supplementary Fig. 18). Furthermore, the ultra-high thermal stability of BPII in −190 to 340 °C is investigated in detail in monodomain BPII (Supplementary Figs. 19–22).

**Direct dynamic track of the double-twisted cylinders' arrangement**. To further investigate the rearrangement of DTCs in BPLCs, direct submicrometer scale real- and reciprocal-space observations of PS-BPLCs were conducted using TEM and syn-SAXS. Figure 2a shows the results for characteristic monophasic BPII$_{\{100\}}$, which appears as a bright bluish domain in the POM image (Fig. 2a$_1$). The TEM images in Fig. 2a$_2$–a$_3$ show typical microscopic SC structures, which are the real-space arrangement of DTCs in BPII$_{\{100\}}$[38] (Supplementary Figs. S23b, S24a, and S27). The fast Fourier transform (FFT) pattern obtained through a TEM analysis (Supplementary Fig. 28a$_2$) exhibits speckles of BPII$_{\{100\}}$ with four-fold symmetry. As the speckles of the FFT pattern transferred from the theoretically predicted model (Supplementary Fig. 28a$_3$) are the same as those observed experimentally (Supplementary Fig. 28a$_2$), the DTC arrangement observed in the TEM image (Fig. 2a$_3$ and Supplementary Fig. 28a$_1$) can be schemed using the theoretical model (Supplementary Figs. 25a, 26, and 28a$_4$). The lattice constant of 171.82 nm for BPII is derived from the relative FFT pattern (Supplementary Fig. 27), corresponding to a value of 167.33 nm from the syn-SAXS pattern (Fig. 2a$_4$ and Supplementary Figs. 46b, 47b, 48a, and 49). Figure 2b shows the characteristic structures of monophasic BPI$_{\{110\}}$, which appear as a green domain in the POM image (Fig. 2b$_1$). Fig. 2b$_2$-b$_3$ (Supplementary Figs. 23a and 29–35) show the BCC structures of polydomain BPI with distinct crystal orientations, and a unit cell is indicated by a yellow dotted rectangle (Fig. 2b$_3$), with a lattice constant of 252.57 nm (labeled *a*) derived from the FFT pattern (Supplementary Fig. 31),

corresponding to a value of 254.48 nm from the syn-SAXS pattern (Fig. 2b$_4$ and Supplementary Fig. 48b). Similarly, the arrangement of DTCs in the TEM images can be described through a theoretical model (Supplementary Figs. 25b, 28b$_4$, and 29). The arrangement of DTCs close to the interfaces between BPII and BPI in the BPIII/BPII/BPI core-shell configuration are investigated in Stage III (Fig. 3a and Supplementary Figs. 36–39), where, a clear interface between bluish BPII$_{\{100\}}$ and green BPI$_{\{110\}}$ appears in the POM image (Fig. 3a$_1$); that is, BPI is surrounded by BPII, forming a BPI core encapsulated by BPII. The interface between the highly ordered BPI$_{\{110\}}$ and BPII$_{\{100\}}$ on the submicrometer scale is indicated by the yellow dashed lines (Fig. 3a$_2$–a$_3$). FFT analyses (Supplementary Fig. 28c$_2$) suggest a combination of sharp diffraction speckles of BPI$_{\{110\}}$ and four-fold symmetric speckles of BPII$_{\{100\}}$, which corresponds to the theoretically predicted FFT pattern (Supplementary Fig. 28c$_3$). The syn-SAXS patterns of the monodomain BPLCs (with hybrid phases of BPI$_{\{110\}}$ and BPII$_{\{100\}}$) exhibit sharp diffraction speckles (Fig. 3a$_4$ and Supplementary Fig. 50), suggesting a highly ordered structure with the same lattice parameters at the interface and bulk region; that is, without a transitional region. The overlapping of speckles diffracted from BPI$_{\{211\}}$ and BPII$_{\{100\}}$ (Supplementary Figs. 46b$_1$–e$_1$, 47b$_1$–d$_1$, 48, and 50), suggesting that the lattice orientation relationships are $\{110\}_{BPII}//\{211\}_{BPI}$ (Supplementary Fig. 52). The crystal lattices near the interfaces between BPI and BPII are coherent (Supplementary Figs. 36, 38, 39a$_1$–a$_2$), where a DTC simultaneously belongs to the BPII and BPI unit cells, proving the diffusionless behavior of DTCs during the phase transition from BPII to BPI. In this case, molecules can diffuse constantly and flows in a three-dimensional (3D) crystalline and their directors are determined by their position in DTCs. The DTCs during the phase transformation of BPI/BPII occur through large and collective reorganization, which do not shape and reform as a whole through diffusion. The coherent crystal lattices provide evidence for the probable thermoelastic martensitic transformation between BPII and BPI. In addition, the rearrangement behaviors of DTCs during the transformation of BPII/BPI are illustrated (Supplementary Figs. 33 and 52–54).

Figure 3b shows the interface between BPIII and BPI$_{\{110\}}$ of BPIII/BPI core-shell configurations in Stage III, indicated by a distinct color change in a crossed POM image obtained in reflection mode (Supplementary Fig. 40d$_1$–d$_2$). A green BPI$_{\{110\}}$ domain appears against a dark BPIII background, exhibiting periodic BPI$_{\{110\}}$ on the left and amorphous BPIII on the right (Fig. 3b$_2$). The real-space observation of the interface between BPIII and BPI$_{\{110\}}$ at the microscale proves the direct transition from BPIII to BPI without the mesophase of BPII, forming BPIII/BPI core-shell configurations. In addition, the arrangement of the DTCs in BPI$_{\{110\}}$ close to the interface (Supplementary Fig. 41b and e) are the same as those in the bulk region (Supplementary Fig. 41c and f). A DTC can be observed across the interface and simultaneously exists both in BPIII and BPI unit cells, confirming the DLPT between BPIII and BPI (see the DTCs highlighted by the curved dotted lines in Supplementary Fig. 39c). This phenomenon directly proves that the DTC reconfigure diffusion-lessly and form phases rather than undergoing a diffusion process first and then reorganization. An FFT analysis (Supplementary Fig. 28d$_2$) suggests that the pattern consists of both diffraction speckles from the periodic BPI$_{\{110\}}$ and a scattering ring from the amorphous of BPIII. The interface between BPII$_{\{100\}}$ and BPIII are observed (Fig. 3c and Supplementary Fig. 29), indicating a submicrometer scale transformation between BPII$_{\{100\}}$ and BPIII. The POM image (Supplementary Fig. 40e$_1$–e$_2$) shows that a bright bluish BPII$_{\{100\}}$ domain appears inside the dark BPIII background. TEM images (Fig. 3c and Supplementary Figs. 28e

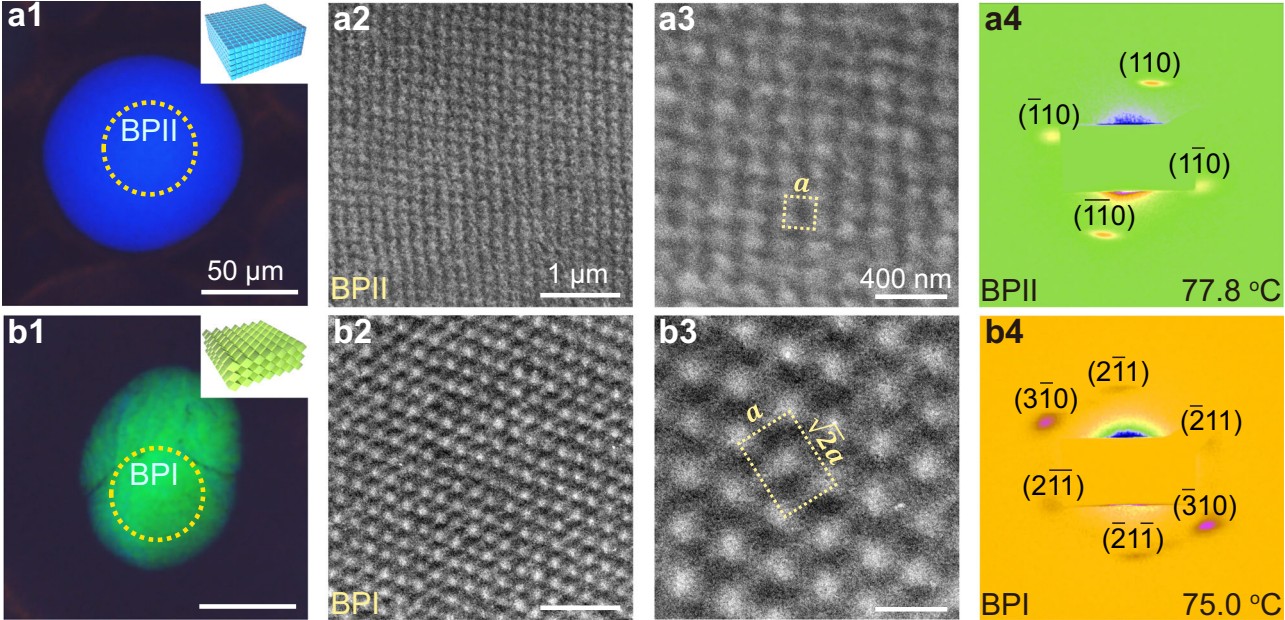

**Fig. 2 TEM images and syn-SAXS analysis of the monophases of BPI and BPII. $a_1$, $b_1$** Textures observed in reflection mode using cross-polarized microscopy. The yellow circle highlights the monophase or interface in core-shell configurations. The inserts in $a_1$, $b_1$ are schematic illustrations of the arrangements of cubic cells at the interfaces of BPIII, BPII, and BPI or monophasic BPII and BPI. TEM images of $a_2$, $a_3$ BPII, $b_2$, $b_3$ BPI. $a_4$, $b_4$ Syn-SAXS patterns with background subtracted from a monodomain $a_4$ BPII$_{\{100\}}$, $b_4$ BPI$_{\{110\}}$.

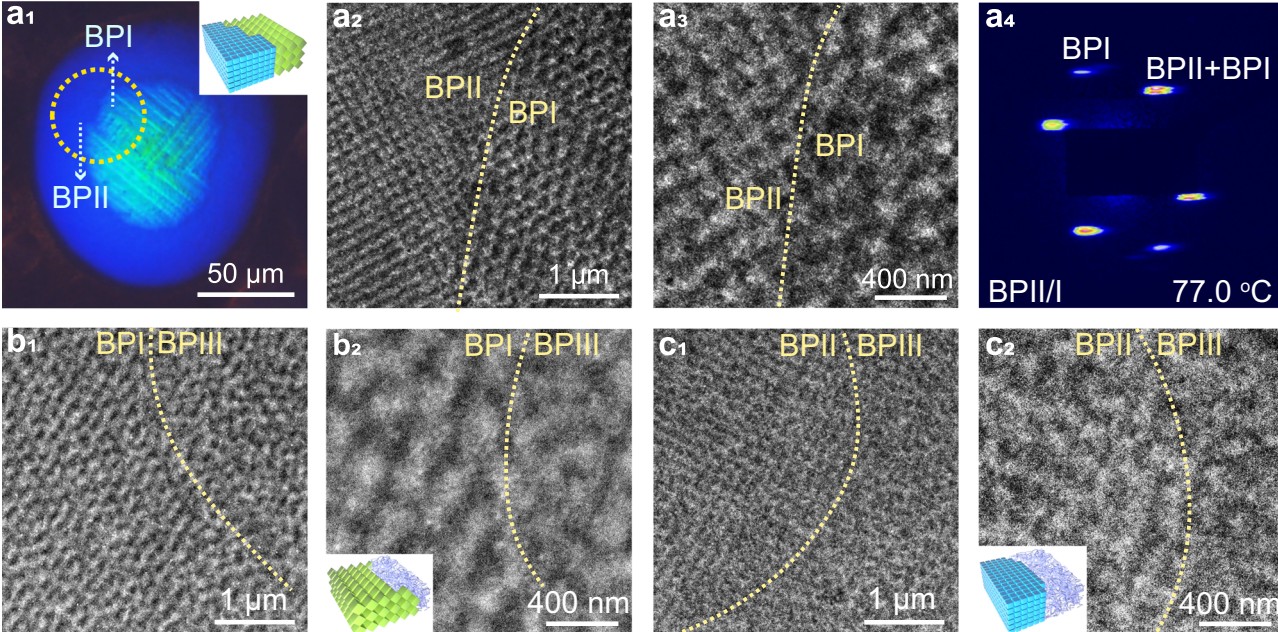

**Fig. 3 TEM images and syn-SAXS analysis of the core-shell configurations of BPI/BPII/BPIII, BPI/BPIII, and BPII/BPIII. $a_1$** Textures observed in reflection mode using cross-polarized microscopy. The yellow circle highlights the monophase or interface in core-shell configurations. The inserts in $a_1$, $b_2$, and $c_2$ are schematic illustrations of the arrangements of cubic cells at the interfaces of BPIII, BPII, and BPI or monophasic BPII and BPI. $a_2$, $a_3$ TEM images of the interface between BPI$_{\{110\}}$ and BPII$_{\{100\}}$, $b_1$, $b_2$ the interface between BPI$_{\{110\}}$ and BPIII, and $c_1$, $c_2$ the interface between BPII$_{\{100\}}$ and BPIII. $a_4$ Syn-SAXS patterns with background subtracted from a monodomain $a_4$ hybrid phase of BPI$_{\{110\}}$ and BPII$_{\{100\}}$. "BPII/I" refers to hybrid phases of BPII and BPI. The crystal lattices of BPI and BPII are coherent and a DTC can simultaneously exist in both BPI and BPII units, confirming that the DTC undergoes a reconfiguration process without diffusion. In addition, a DTC is observed across the interface of BPIII/BPI or BPIII/BPII, indicating that the DTCs do not diffuse when phase transition occurs from BPIII to BPII or BPI. The interfaces of BPI/BPII, BPII/BPIII, and BPI/BPIII are clear without a nearby transitional region. Based on the syn-SAXS results in reciprocal-space, the crystal orientation relationship between BPI and BPII is confirmed as $\{211\}_{BPI}//\{110\}_{BPII}$. Moreover, a perfect monodomain BPI with a large size is characterized.

and 42) show a clear interface between the highly ordered structures with four-fold symmetric (BPII$_{\{100\}}$) and amorphous structures (BPIII). The theoretically predicted FFT pattern (Supplementary Fig. 28e$_3$) matches well with the FFT pattern transfer from the TEM results (Fig. 3c$_2$ and Supplementary Fig. 28e$_2$), suggesting that the arrangement of DTCs in the TEM image (Supplementary Fig. 28e$_1$) can be represented by the theoretical model (Supplementary Fig. 28e$_4$). Here, a phenomenon is observed similar to that of the BPII/BPIII interface, in which a DTC can cross the interface and simultaneously exist in both BPII unit cells and BPIII (Supplementary Fig. 39b). Thus, the BPIII-to-BPII phase transition is also a DLPT process.

**Investigation of the lattice orientation by angle-resolved microspectroscopy.** BPI and BPII as soft artificial three-dimensional photonic crystals[26,41–43], show promise for optical applications owing to their photonic band structures[54,55]. To further understand the optical properties of the DLPT process of BPLCs, ARM was used to characterize the two-dimensional transmittance/reflectance spectra (see Supplementary Figs. 51 for the working principle) of BPLC for obtaining the projected band structures. Figure 4a shows the in situ dynamic track of the phase transition of polydomain BPLCs using ARM (Supplementary Figs. 52 and 53). The measured reflectance (transmittance) spectra are plotted as a function of the light frequency $\omega$ and the in-plane wave vector $\mathbf{k}_{\parallel}$, where $|(\mathbf{k} + \mathbf{G})_{\parallel}| = (\omega/c)\sin\theta_{\mathrm{in}}$[56] (G, reciprocal-lattice vector; $c$, light speed; $\theta_{\mathrm{in}}$, incident depression angle). The Bloch wave vectors are labeled $\Gamma$, $H$, $N$, and $P$ for BCC BPI (Fig. 4b$_5$) and $R$, $X$, $S$, and $\Gamma$ for SC BPII (Fig. 4c$_3$) according to the high-symmetry directions of the Brillouin zones (BZs)[54,55]. The bright streaks in these plots (Fig. 4a$_1$–a$_5$ and Supplementary Figs. 52 and 53) are extensions of the features of the reflection spectra in Fig. 1a$_4$–e$_4$, which contains information on the Bragg scattering processes. Herein, no streak (i.e., no reflection signal) is observed in Stage I (Figs. 1a$_4$ and 4a$_1$) for BPIII of an amorphous structure. A bright reflection streak appears in Figs. 1b$_4$ and 4a$_2$ where the frequency of the peak near the normal incident angle ($f_{\mathrm{dip}}$) is ~637 THz ($\lambda = 463$ nm, normalized frequency ($f_{\mathrm{nor}}$) = 0.36, blue; see "Methods" section for details), indicating the emergence of BPII$_{\{100\}}$ in Stage II. After the temperature drops, a weak reflection streak with $f_{\mathrm{dip}} \approx 554$ THz ($\lambda = 541$ nm, $f_{\mathrm{nor}} =$ 0.47, green) was observed in Stage III (Fig. 4a$_3$), which corresponds to a preliminary stage of the phase transition from BPII$_{\{100\}}$ to BPI$_{\{110\}}$, in accordance with the appearance of two peaks in Fig. 1c$_4$. Subsequently, two apparent streaks of BPI$_{\{110\}}$ and BPII$_{\{100\}}$ were observed from ARM results (Fig. 4a$_4$). The intensity of the BPI$_{\{110\}}$ streak increased and that of BPII$_{\{100\}}$ decreased, as shown in Fig. 4a$_3$–a$_4$ (from Stages III to IV), respectively. In addition, both BPII$_{\{100\}}$ and BPI$_{\{110\}}$ streaks are clear, indicating a sudden reconfiguration of unit cells near the interface without a transitional region, which may cause a broadening and blurring of the streaks. Finally, only one strong streak representing BPI$_{\{100\}}$ is observed in Stage V (Figs. 1e$_4$ and 4a$_5$). Owing to the specific reflection streak for a particular crystal orientation, in situ ARM can be used to dynamically track the martensitic transformation from BPII to BPI in a core-shell configuration and BPI nucleating at the center of BPII (Supplementary Figs. 54 and 55).

To distinguish the shapes and positions of the BPI$_{\{110\}}$ and BPII$_{\{100\}}$ signals in Fig. 4a, ARM is applied to polymer-stabilized monodomain BPI$_{\{110\}}$ (Fig. 4b$_2$–b$_4$) and BPII$_{\{100\}}$ (Fig. 4c$_2$). The preparation and phase transformation of monodomain BPI$_{\{110\}}$ and BPII$_{\{100\}}$ are shown in supplementary Movie 3. Each plot was measured at a certain azimuthal angle. We can see a regular dark streak for each phase, similar to that observed in the in situ

measurements. For BPI$_{\{110\}}$ (Fig. 4b$_2$–b$_4$), the transmittance peak at the normal angle ($f_{nor}$) appears at ~0.49 ($\lambda = 520$ nm, green), where the anticrossing shifts with varying azimuthal angles. By contrast, $f_{nor}$ is ~0.36 ($\lambda = 463$ nm) for BPII$_{\{100\}}$, which has a constant crescent shape (Fig. 4c$_2$) and appears as a monophasic blue (Fig. 4c$_1$). Therefore, the two reflection streaks in the ARM spectra of polydomain PS-BPLCs (Fig. 4a and Supplementary Figs. 52 and 53) can be clearly distinguished by their positions and shapes. Thus, the crystal orientation during the phase transformation between BPI$_{\{110\}}$ and BPII$_{\{100\}}$ can be determined by dynamic tracking of ARM (Fig. 4a). It is observed that the dips of BPI$_{\{110\}}$ and BPII$_{\{100\}}$ streaks match with each other (Supplementary Figs. 52, 53, and 56c$_1$) during the early stage (-Stage III) of phase transformation (Fig. 4a$_3$), exhibiting BPI$_{\{110\}}$ // BPII$_{\{100\}}$. In Stage IV, the orientation relationship of $\{211\}_{BPI}$// $\{110\}_{BPII}$ is constructed and a small angle of 9.74° (=arccos$\frac{\sqrt{3}}{3}$-45°) are observed between $\{110\}_{BPI}$ and $\{100\}_{BPII}$ (Supplementary Fig. 56a$_1$–a$_2$). The relationship of $\{211\}_{BPI}$//$\{110\}_{BPII}$ was proven by the overlapping of the speckles in the syn-SAXS pattern (Supplementary Figs. 46b$_3$–e$_3$, and 47c$_1$–d$_1$, and 56d$_1$–d$_4$) and a similar $q$ value (Supplementary Fig. 48), causing a deviation of $\{110\}_{BPI}$ //$\{100\}_{BPII}$ (Supplementary Fig. 56b$_1$–b$_2$). The deviation of $\{110\}_{BPI}$ //$\{100\}_{BPII}$ with an angle of 9.74° existed is not only predicted by the theoretical models (Supplementary Fig. 56a$_1$–a$_3$) but also confirmed experimentally by Syn-SAXS and ARM (Supplementary Fig. 56c, d). It can be seen from ARM results that the dip of BPI$_{\{110\}}$ streak does not change, whereas that of BPII$_{\{100\}}$ does, which means that the rotations for 9.74° of the parent lattice (BPII$_{\{100\}}$) are required during the thermoelastic martensitic transformation.

Furthermore, the azimuthal angle and specific crystal orientation of BPLCs can be determined by the relationship between BPI$_{\{110\}}$ and the rubbing direction of the LC cell based on a simple optical characterization of ARM (Fig. 4b$_1$ and Supplementary Fig. 62). In this study, the anticrossings on the streak patterns of BPI$_{\{110\}}$ move inward and outward periodically by changing the azimuthal incident angle of the light (Fig. 4b$_2$–a$_4$ and Supplementary Figs. 57, 60, and 63), whereas the streaks of ARM spectra showed a constant crescent shape for BPII$_{\{100\}}$ under the same conditions (Fig. 4c$_2$ and Supplementary Figs. 58 and 61) owing to the distinct crystal symmetry of BPI and BPII, in which the distance between the edges of the BZ and normal incident angle (Supplementary Fig. 64) was longer in BPII than that in BPI. According to the direct relationships between the shapes of directional streaks and azimuthal angles, the BPI with a certain azimuthal angle exhibits a particular shape of streak in two-dimensional transmittance/reflectance spectra, including the positions of the anticrossings and the direction of the in-plane wave vector (Fig. 4b$_2$–b$_4$).

Combining the TEM image of the BPI$_{\{110\}}$ (Supplementary Fig. 32), the shape of the streaks indicates that the normal incident angle corresponds to the $\Gamma$-$N$ direction ($\overline{\Gamma N}$), i.e., [110] direction (Fig. 4b$_1$ and Supplementary Fig. 65). One maximum incident angle is observed where anticrossings appear at the normalized wave vector $\mathbf{k}a/2\pi \approx \pm 0.50$ ($\theta_{in} \approx \pm 58°$) (Fig. 4b$_2$ and Supplementary Fig. 66a), and the azimuthal angle corresponds to $\overline{NP}$. The minimum incident angle is obtained at the non-high-symmetry point $L_{\perp}$ (the perpendicular foot of $\Gamma$ on $\overline{HP}$) where the anticrossings are located at $\mathbf{k}a/2\pi \approx \pm 0.41$ ($\theta_{in} \approx \pm 50°$) (Fig. 4b$_3$ and Supplementary Fig. 66b). The other maximum incident angle is observed where the angles of the anticrossings exceed the measurement range, with the azimuthal angle in the $\overline{NH}$ direction (Fig. 4b$_4$). Thus, the given crystallographic orientations and rubbing direction are indicated by the red lines and yellow arrows in the POM image, respectively (Fig. 4b$_1$); the angle between

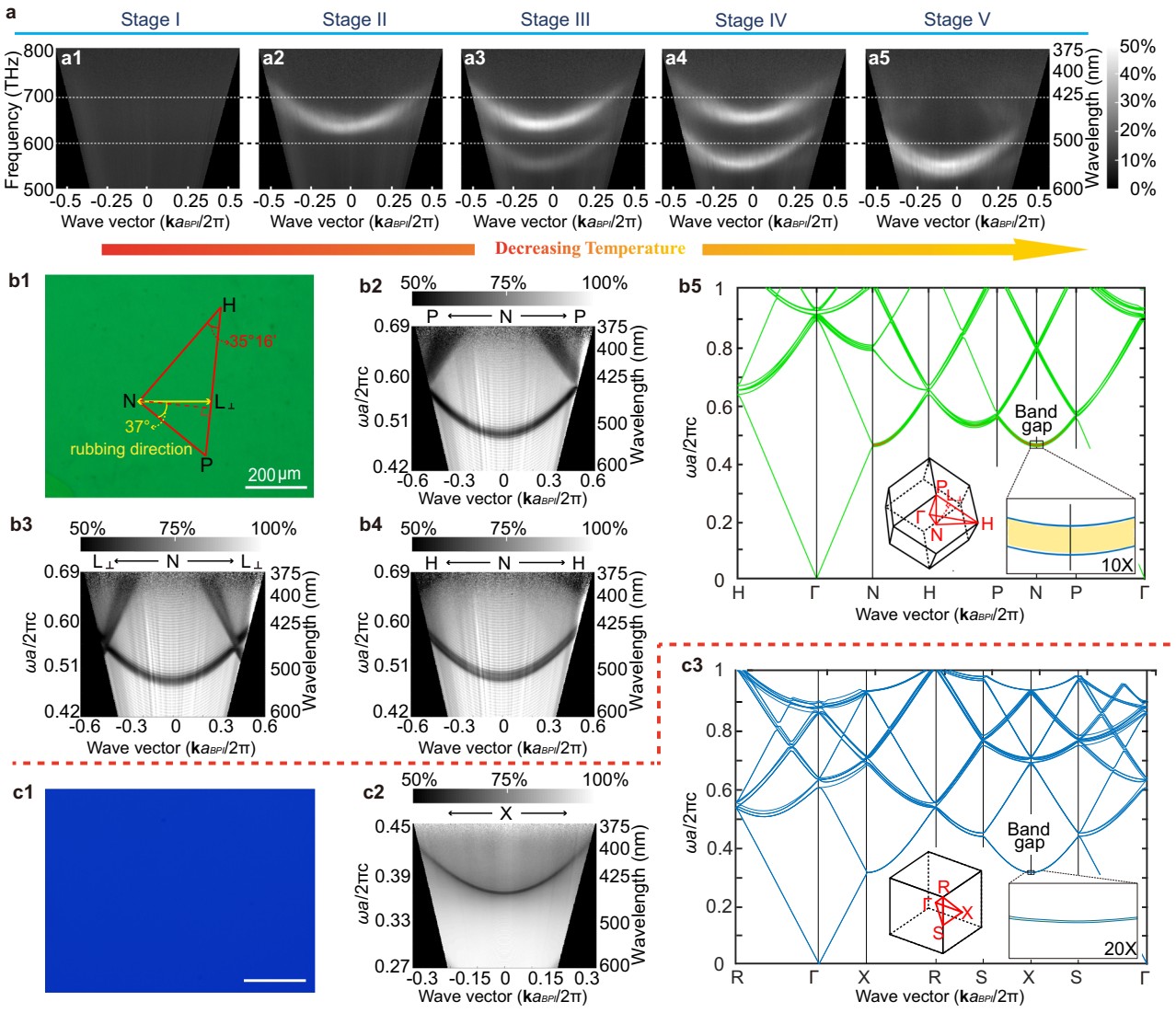

**Fig. 4 Relationships of lattice orientation during DLPT characterized by ARM and simulated band structures. a** In situ observation of micro-reflectance streaks of polydomain BPLCs during the phase-transition process from Stages I to V. The wave vectors are normalized by $\frac{ka}{2\pi}$, where $a = 254.48$ nm is the lattice constant of BPI. The color scale indicates the reflection of the measured area. The color scale besides **a₅** is also available for **a₁–a₄**. **b₁** The typical texture of monodomain BPI$_{\{110\}}$ (with $\lambda_c = 520$ nm at normal incidence after polymerization) obtained using POM. The red line and yellow arrow represent the detection directions and rubbing direction, respectively. **b₂**, **b₃**, and **b₄** Contour plots of measured transmittance of BPI in the $N{\to}P$, $N{\to}L_\perp$ and $N{\to}H$ directions, respectively. **b₅** Calculated photonic band structure on the edges of irreducible BZ of BCC BPI. **c₁** The typical texture of monodomain BPII$_{\{100\}}$ (with $\lambda_c = 463$ nm at normal incidence after polymerization) obtained using POM. **c₂** Contour plots of measured transmittance of BPII. **c₃** Calculated photonic band structure on the edges of the irreducible BZ of BPII. Inserts in **b₅**, **c₃** show BZ of BPI or BPII with symmetry points and opening of the local bandgap. Three closely spaced pseudogaps are observed along the $\overline{\Gamma N}$ ([110]), $\overline{\Gamma P}$ ([200]), and $\overline{\Gamma H}$ ([210]) directions of BPI and the $\overline{\Gamma X}$ ([100]), $\overline{\Gamma S}$ ([110]) and $\overline{\Gamma R}$ ([111]) directions of BPII. The BPI covered by BPII in the BPIII/BPII/BPI core-shell configurations is confirmed through in situ ARM, and the topotaxial relationship between BPI and BPII in Stage III is found to be $\{100\}_{\mathrm{BPII}}//\{110\}_{\mathrm{BPI}}$. Significantly, a rotation of the parent phase (BPII) occurs during cooling from Stage III to Stage IV, resulting in BPI$_{\{211\}}//$BPII$_{\{110\}}$ and an ~$\{100\}_{\mathrm{BPII}}//\{110\}_{\mathrm{BPI}}$ in Stage IV. In addition, the azimuthal angle between the [100] direction and rubbing direction is determined by angle-resolved reflectance microspectroscopy through the [110] direction.

[001] and rubbing directions was measured as 37° (Fig. 4b₁ and Supplementary Fig. 59). In addition, based on the TEM image of the BPII$_{\{110\}}$, the shape of the streaks indicates that the normal incident angle of the BPII sample should be along the $\overline{\Gamma X}$ ([100]) direction (Supplementary Fig. 65).

To understand the relationship between the measured ARM spectra and the band structures of BPI$_{\{110\}}$ (along the $\overline{\Gamma N}$ ([110]) direction) and BPII$_{\{100\}}$ (along the $\overline{\Gamma X}$([100]) direction), we simulated the stopband of the BPLCs along with the high-symmetry directions of the irreducible BZs using the plane-wave expansion method, as shown in Figs. 4b₅ (BPI) and 4c₃ (BPII).

Directional bandgaps appear on the entire edge of the irreducible BZs of BPII and BPI. Several closely spaced pseudogaps were obtained, such as BPI along the $\overline{\Gamma N}$ ([110]), $\overline{\Gamma P}$ ([200]), and $\overline{\Gamma H}$ ([210]) directions, and BPII along the $\overline{\Gamma X}$ ([100]), $\overline{\Gamma S}$ ([110]), and $\overline{\Gamma R}$ ([111]) directions. The bandgap of BPII along the $\overline{\Gamma X}$ ([100]) direction (~$0.0003\ \omega a_{\mathrm{BPII}}/2\pi c$) is narrower than that of BPI along the $\overline{\Gamma N}$ ([110]) direction (~$0.0096\ \omega a_{\mathrm{BPI}}/2\pi c$); consequently, the full width at half maximum (FWHM) of the streaks in the transmittance/reflectance spectra is smaller (Supplementary Fig. 67), which might be useful for improving the sensitivity of the sensor and realizing dual-wavelength lasers[44]. The calculated

anticrossing shape of BPI and the crescent shape of BPII are in accord with the experimental results. As for BPI, the calculated positions of the BZ high-symmetry points ($\overline{\mathrm{NP}} = \frac{1}{2}\frac{2\pi}{a_{\mathrm{BPI}}} = 0.5\frac{2\pi}{a_{\mathrm{BPI}}}$, $\overline{\mathrm{NL}}_\perp = \frac{\sqrt{6}}{6}\frac{2\pi}{a_{\mathrm{BPI}}} \cong 0.408\frac{2\pi}{a_{\mathrm{BPI}}}$, as shown in Supplementary Fig. 64) agree well with the measured values (Fig. $4b_2$–$b_3$). In addition, the calculated normalized frequencies of the bandgaps of the BPI (0.46) and BPII (0.31) (for the normal angle) case are consistent with the ARM results and measured lattice constants.

**Difussionless transformation of BPII-to-BPI, BPIII-to-BPI, and BPIII-to-BPII.** BPI is considered to be a thermoelastic martensitic transformation for the following reasons: (i) A shear termed as an invariant plane strain is required[57] to be parallel to the habit plane. The transition between BPII and BPI is a result of the strain release through twinning (Supplementary Fig. $12a_1$–$b_2$). (ii) Twins in martensite may be self-accommodating and the energy is reduced by the surface reliefs[57]. The surface relief in the BPLCs is observed in both poly- and single-domain BPI during a phase transformation from BPII (Supplementary Fig. $12c_1$–$c_2$). (iii) The lattice orientation relationships are proven to be $\{110\}_{\mathrm{BPII}}//\{211\}_{\mathrm{BPI}}$ (Fig. 3 and Supplementary Figs. $47c_1$–$d_1$, and $56d_2$) and $\sim\{100\}_{\mathrm{BPII}}//\{110\}_{\mathrm{BPI}}$ (9.74° between $\{100\}_{\mathrm{BPII}}$ and $\{110\}_{\mathrm{BPI}}$) (Supplementary Figs. 12a and 56b-c). (iv) The crystal lattices of BPII and BPI are coherent at the interface, which is proven by the similar $q$ values corresponding to $\{110\}_{\mathrm{BPII}}$ and $\{211\}_{\mathrm{BPI}}$ (Supplementary Figs. $46b_3$–$e_3$, $47c_1$–$d_1$, and 48) and the TEM images at the interface between $\{100\}_{\mathrm{BPII}}$ and $\{110\}_{\mathrm{BPI}}$ (Supplementary Figs. 36, 37, 38, and 39a), in which a DTC simultaneously belongs to BPII and BPI units, confirming that the DTCs do not diffuse in the phase transformation of BPI/BPII. (v) At a certain temperature, the martensitic nucleus of BPI is formed in the center of BPII, in which the formed martensite BPI domains continue to grow and thicken with a further decrease in temperature; that is, BPI can grow at variable temperatures (Fig. $1c_2$–$e_2$ and Supplementary Figs. 1, 10, and 11). (vi) The forward and backward martensitic transformations between BPII and BPI are rapid and completely reversible for >50 conversion-reversion cycles (Supplementary Figs. 10a, 11 and Movie 2). The reverse transformation occurs not through re-nucleation, but through a gradual reduction of the existing BPI domains, which appears to be a reversal of the forward transformation. (vii) The phase transformation between BPI and BPII is rapid with little thermal hysteresis (Supplementary Figs. 10a, 11 and Movie 2), proving that the phase-transition process is always an equilibrium process at different temperatures. (viii) Both the BPII (SC) and BPI (BCC) composed of DTCs have an ordered structure, and the macroscopic volume change caused by the phase transformation is almost neglected.

Furthermore, the thermoelastic martensitic transformation between BPII and BPI produces a completely reversible switching of the texture at high and low temperatures (Supplementary Figs. 10 and 11 and $15a_1$, $b_1$), which can be analogous to the shape-memory effect in a solid system. Figures 5–6 and Supplementary Fig. 15 show the application of temperature-switchable quick response (QR) codes at high and low temperatures. The switchable QR code can be used as an "ID card" for the anti-counterfeiting of medicines, luxury wines, wristwatches, and jewelry because it offers greater security than a normal static QR code.

In addition, it is confirmed that the phase transformations of BPIII-to-BPII and BPIII-to-BPI are also DLPT for the following reasons: (i) during the unusual phase transformation from BPIII to BPI (Supplementary Figs. 8e, f and 9), or from BPIII to BPII (Supplementary Figs. 5–7), a layer of BPIII exists between the

isotropic state and BPI (Fig. $1c_2$ and Supplementary Figs. 4 and 8–10) or BPII (Fig. $1d_2$ and Supplementary Figs. 7f and 8f, g), proving that the formation of BPII or BPI requires pre-formed DTCs of BPIII. BPII or BPI cannot nucleate directly from the isotropic state. (ii) DTCs were observed across the interface of BPIII/BPII (Supplementary Fig. 39b) or BPIII/BPI (Supplementary Fig. $39c_1$–$c_2$). A DTC is observed simultaneously in both BPII units and BPIII or both BPI units and BPIII crossing the interfaces. (iii) No transition area of diffused DTC (area of molecules with random arrangement) is observed at the interface of BPII/BPIII or BPI/BPIII (Fig. 3b–c and Supplementary Figs. 28d–e, 41, and 42). Thus, although BPIII is amorphous, the DTCs are structural units for BPI, BPII, and BPIII, which are organized from molecules that do not diffuse during the phase-transition process.

In conclusion, all of the phase-transition processes in BPLCs (BPIII-to-BPII, BPIII-to-BPI, and BPII-to-BPI) are diffusionless.

Most of the reported DLPT processes occur in atomic crystals[57,58], and thus investigations into the DLPT behavior in the soft matter such as BPLCs are recalling. The common martensitic transformation only has one step; for example, the transition from γ-Fe to α′-Fe or isomerization from α- to β-$Cd_{37}S_{21}$[59]. Specifically, two types of successive DLPT were confirmed in our experiment when the BPLCs moved from Stage I to V: (i) a two-step process consisting of BPIII-to-BPII and BPII-to-BPI sequentially, resulting in sequential transitions from BPIII to BPIII/BPII and BPIII/BPII/BPI core-shell configurations and finally to BPI (Supplementary Fig. 8a–d). (ii) a three-step process consisting of BPIII-to-BPI, BPIII-to-BPII, and BPII-to-BPI sequentially, causing sequential transitions from the BPIII domains to BPIII/BPI and BPIII/BPII/BPI core-shell configurations and finally to BPI (Supplementary Fig. 8e–h).

In addition, the DLPT process of BPLCs undergoes five stages during the cooling process with core-shell configurations, which have been developed for unique applications such as fabrication of BPLCs with ultra-large domain sizes (Supplementary Figs. 13 and 14), micro-area lasing (Fig. $7c_1$), responsive phase transformation, and temperature-switchable binary or ternary QR code (Figs. 5–6 and Supplementary Fig. 15).

**Binary and ternary temperature-switchable QR Codes.** The randomly distributed textures of polydomain BPLCs with large domain sizes are unique and can be applied as binary and ternary QR codes, as shown in Figs. 5 and 6. Here, the binary (Fig. $5a_4$) and ternary (Fig. $5a_5$) QR codes (with a large pixel resolution of $37 \times 37$) were constructed from the POM image (Fig. $5a_1$) experienced the processes of gray-level conversion (Fig. $5a_2$ and $b_2$), contrast change (Fig. $5a_3$ and $b_3$), pixelation (Fig. $5a_4$ and $b_4$), and encoding (Fig. $5a_5$ and $b_5$). The conversion is carried out mainly based on the brightness of the POM image, and each domain (Fig. $5a_1$) can be regarded as a pixel of the QR code. For example, a binary code can be obtained by encoding the "white" or "black" pixels into "1" or "0" and the codes can be further associated with the other codes such as ASCII binary codes. Similarly, a ternary QR code (Fig. $5a_5$–$b_5$) is constructed based on the random distribution of black, blue, and green domains in the POM image (Fig. $5a_1$) owing to the random distribution of the BPIII/BPII/BPI core-shell configurations in the black isotropic background. The ternary QR code contains high-density information: encoding the green pixels originated from the domains of $BPI_{\{110\}}$, the blue pixels from those of $BPII_{\{100\}}$, and the black pixels from the isotropic background into "2", "1", and "0" (Fig. $5b_5$), respectively.

The DLPT process of BPLC produces a pair of temperature-switchable QR codes (Fig. 6). Followed by heating the hybrid LCs

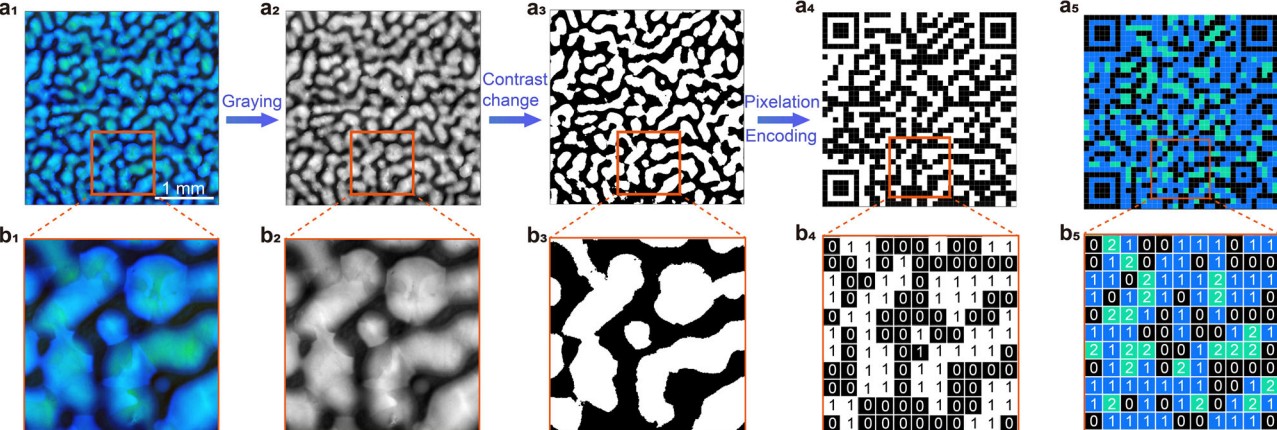

**Fig. 5 Binary anti-faking QR codes. a₁** POM image of PS-BPLCs at Stage IV with low magnification. **a₂** The image obtained after the gray-level conversion of **a₁**. **a₃** An image of the binarization of **a₂**. **a₄** The binary codes obtained from **a₃** after pixelation and encoding processes. **a₅** Large-scale ternary QR code with a pixel resolution of 37 × 37 encoded from **a₁** undergoing ternary and pixelation processes. **b₁–b₃** The magnified image of the orange squared area in **a₁–a₃**. **b₄–b₅** The binary and ternary codes of the orange-squared area in **a₄**, **a₅**. In binary codes, the "black" or "white" pixel represents "0" or "1". In ternary codes, the "green", "blue", or "black" pixels in **a₅**, **b₅** represents "2", "1", or "0".

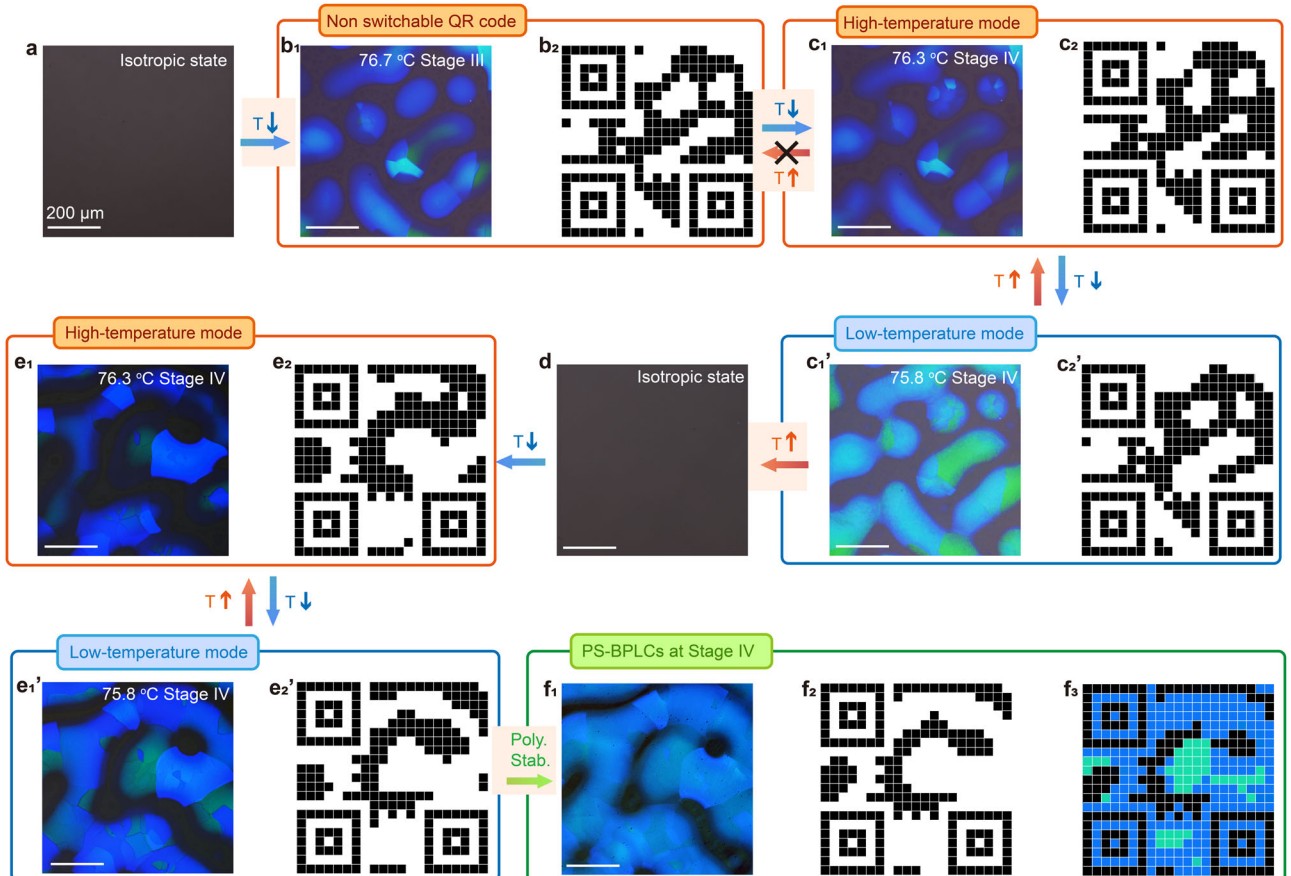

**Fig. 6 Ternary anti-faking temperature-switchable QR codes. a** The initial isotropic state. **b₁** The randomly distributed BPIII domains and configurations of BPIII/BPII, BPIII/BPI, and BPIII/BPII/BPI in Stage III are obtained at 76.7 °C. **b₂** The corresponding binary QR code transferred from **b₁**. **c** The BPIII/BPII/BPI core-shell configurations in Stage IV obtained at **c₁** 76.3 °C and **c₁'** 75.8 °C in the first cooling process. **c₂**, **c₂'** The corresponding binary QR codes transferred from **c₁**, **c₁'**. **d** The isotropic state in the second cooling process. **e₁**, **e₁'** The BPIII/BPII/BPI core-shell configurations in Stage IV observed at **e₁** 76.3 °C and **e₁'** 75.8 °C. **e₂**, **e₂'** The corresponding binary QR codes transferred from **e₁**, **e₁'**. **f₁** The BPIII/BPII/BPI core-shell configurations with high thermal stability by polymer-stabilizing the textures in low-temperature mode **e₂**. **f₂–f₃** The binary and ternary QR codes transferred from the POM image **f₁** using the same methods shown in Fig. 5. "Poly. Stab." refers to "polymer stabilization".

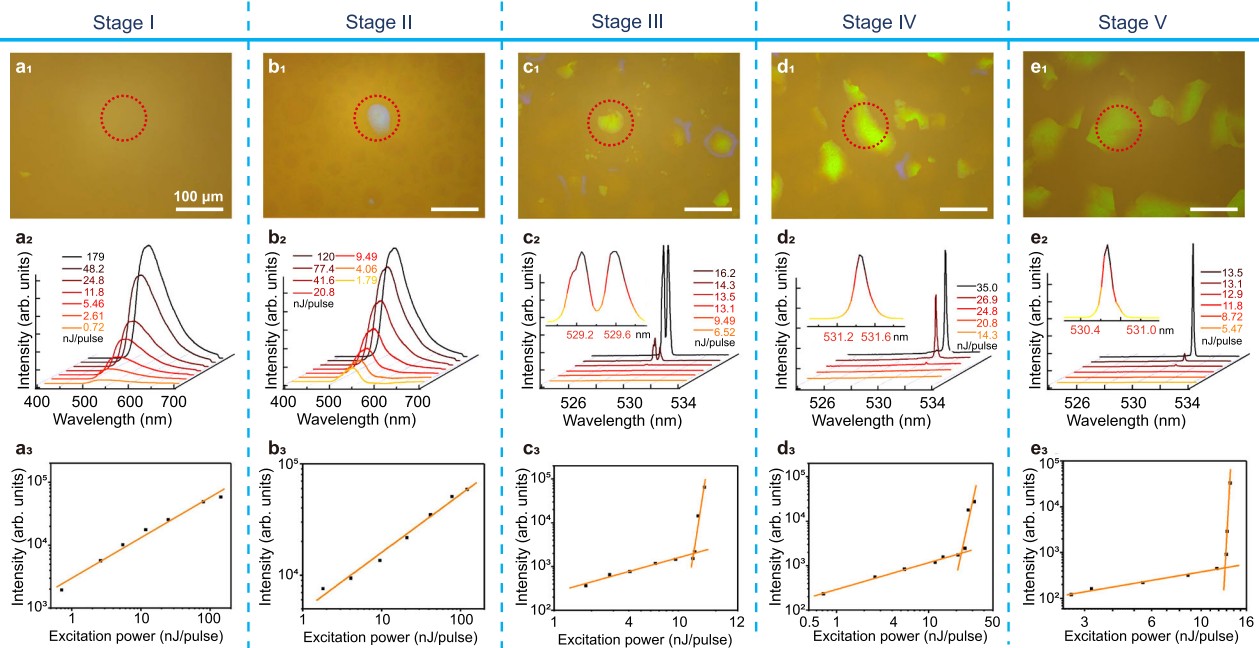

**Fig. 7 Band-edge lasing of BPI in each phase-transition stage. $a_1$–$e_1$** Platelet textures of dye-doped PS-BPLCs captured using an inverted optical microscope in reflection mode; red dotted circles mark the greenish lasing phase platelets. **$a_2$–$e_2$** Emission spectra plotted relative to the low-energy photonic band-edge. The insets show magnified images of the lasing peaks. The measured FWHMs of the lasing peaks vary from 0.579 to 0.105 nm. **$a_3$–$e_3$** Best fits of typical input-output curves for the data below and above the excitation threshold. Lasing is realized in a relatively small BPI domain (~60 μm) based on the high degree of order of BCC structures. The gradual narrowing of the lasing peaks from Stages III to V suggests an improvement in the lattice order of BPI.

to the isotropic state (Fig. 6a), the sample is cooled to Stage III (76.7 °C) at 0.05 °C/min, forming a texture consisting of the randomly distributed BPIII domains and core-shell configurations of BPIII/BPII, BPIII/BPI, and BPIII/BPII/BPI (Fig. 6$b_1$), contributing to a QR code (Fig. 5$b_2$). When decreasing the temperature of the sample to Stage IV (76.3 °C), a texture consisting of BPIII/BPII/BPI core-shell configurations is formed and contributes to a distinctive QR code (Fig. 6$c_2$). Notably, the QR code in Fig. 6$c_2$ cannot recover to that in Fig. 6$b_2$ by increasing the temperature because the phase transformation of BPIII-to-BPII is not completely reversible with large thermal hysteresis. By contrast, a temperature-reversible switching of textures (QR codes) can be obtained at Stage IV between high- and low-temperature modes (Fig. 6c) owing to the thermoelastic martensitic transformation between BPII and BPI. These temperature-switchable textures in soft BPLCs can be considered analogous to the shape-memory in a solid-solid crystal transformation. In detail, the BPI domains grow fast and lead to an increase in brightness and color change of the POM image (Fig. 6$c_1$') during the cooling process from 76.3 °C to 75.8 °C. In comparison, a thermoelastic martensitic transformation from BPI to BPII occurs by a gradual reduction of the existing BPI domain when heating the sample from 75.8 to 76.3 °C. Finally, almost all BPI areas are converted into BPII and the distribution of core-shell configurations is recovered to the original high-temperature mode (Fig. 6$c_1$–$c_2$), forming a QR code similar to the initial state. This reversible temperature-induced phase transformation contributes to a pair of switched QR codes with high duration stability without obvious changes after >50 temperature-change cycles (Supplementary Movie 2). Interestingly, the QR code can be reset by heating the hybrid LCs to the isotropic state (Fig. 6d). This process erases all of the textures. A pair of distinctive temperature-switchable QR codes can be obtained by cooling the hybrid LCs to Stage IV (76.3 °C) for a second cooling process

(Fig. 6e). The texture distribution and the corresponding QR codes (Fig. 6e) are distinct from those of the first cooling process (Fig. 6c), ensuring the uniqueness of the QR codes. The QR code can be further polymer-stabilized to improve the thermal stability and used within a wide temperature range of −190 to 340 °C for specific scenarios such as anti-faking (Fig. 6f).

**Dual-stage thermoelastic martensitic transformation.** To the best of our knowledge, it is difficult to experimentally study the DLPT mechanism based on atomic crystals because of its fast formation speed and complicated characterization technology[57]. Thus, the results of DLPT of BPLCs, which are analogous to atomic crystals, may be good candidates for use in explaining the mechanism of DLPT and provide a possible route for its research. Herein, to further clarify the DLPT mechanism, the band-edge lasing of the BPLCs was investigated (Fig. 7) by doping BPLCs with 1.0 wt% coumarin 6 (C6) as a gain medium. The low-energy edge of the photonic bandgap is designed to overlap the emission spectrum of C6 ($\lambda_{max}$ = 525 nm, λ range of 500–550 nm). The optical textures of the C6-doped BPLCs (C6-BPLCs) (Fig. 7$a_1$–$e_1$) were captured using an inverted optical microscope in reflection mode. The phase-transition temperatures and core-shell config-uration of the C6-BPLCs (Fig. 7$c_1$–$e_1$) are clearly similar to those of the pristine BPLCs (Fig. 1$a_1$–$e_1$). Figure 7$a_2$–$e_2$ show the emission spectra of the C6-BPLCs in terms of the excitation energy supplied by the pump laser (Supplementary Fig. 68). In Stages I and II the emission spectra show only a broad fluorescent peak resulting from spontaneous emission, which can be attrib-uted to the lack of a bandgap effect (BPIII, Supplementary Fig. 69b) or mismatching between the fluorescent emission peak and the bandgap (BPII, Supplementary Fig. 69c). In Stage III, multimode lasing is observed at 529.4 nm (Fig. 7c) for a BPI domain with a size of ~60 μm owing to the highly ordered

periodic structures observed in the TEM image (Supplementary Fig. 69g), where BPI nucleates in the center of BPII and BPIII domains. The lasing achieved in such a small cavity (~60 μm) have the potential to be used in high-brightness and high-saturation displays[60,61]. The thermoelastic martensitic transformation between BPII and BPI with coherent crystal lattices near phase interfaces, is considered to be fundamental to the achievement of distributed feedback surface-emitting micro-laser arrays in PS-BPLCs with a high integration, which is promising for next-generation optical lasing devices.

Single-mode lasing was realized at 531.43 nm (Fig. 7d, Stage IV), 530.70 nm (Supplementary Fig. 70), and 530.67 nm (Fig. 7e, Stage V). The lasing wavelength is modified by the blue-shift of the reflectance spectra from 523.7 (Stage III) to 522.9 (Stage IV) and 518.9 nm (Stage V) (Supplementary Figs. 69a–f), which is consistent with the slight change in crystal size in the TEM images (Supplementary Fig. 69g–i). The FWHMs of these measured lasing signals were 0.579, 0.172, and 0.105 nm for Stages III, IV, and V, respectively. The decrease in the FWHM is attributed to the improved order of these soft photonic microcavities, which can contribute to an increase in the Q-factor. The thresholds are obtained from the knee of the input-output curves (Fig. 7a₃–e₃), which vary from 10.32 to 22.44 and 12.86 nJ/pulse in Stages III, IV, and V, respectively. Consequently, the Q-factors are 914, 3090, and 5054 for the lasing peaks in Stages III, IV, and V, respectively. The enhanced lasing quality during the phase transformation can be attributed to the improvement in crystal quality from Stages III to V, as confirmed by the reflectivity (Supplementary Fig. 69d–f) and TEM images (Supplementary Fig. 69g–i). Therefore, the quality of the crystal lattice gradually improves with the thermoelastic martensitic transformation from SC BPII to BCC BPI. Owing to the evolution of lasing performance in BPI, it was found that the thermoelastic martensitic transformation is a dual-stage process, which is generally difficult to characterize in metals, alloys, or ceramic materials owing to their complex conditions. During this process, the fresh-formed crystalline grains exhibit a high degree of order, and the quality of the crystal lattice can be further improved during the phase-transition process.

## Conclusions

In summary, the DLPT processes of BPIII-to-BPI, BPIII-to-BPII, and thermoelastic martensitic phase transformation of BPII-to-BPI were dynamically tracked and directly characterized in sub-micrometer real-space using POM, TEM, and ARM. In particular, DTCs in BPII→BPI show a diffusionless, collective, and highly coordinated motion, which is a hallmark of a reversible thermoelastic martensitic transformation. Besides, diffusionless behaviors of DTCs are also proven in BPIII-to-BPI, BPIII-to-BPII. In addition, three types of core-shell configurations are formed: BPIII/BPI, BPIII/BPII, and BPIII/BPII/BPI. Several applications based on the core-shell configurations have been achieved such as temperature-switchable binary and ternary QR codes, micro-area lasing, and fabrication of BPLCs with ultra-large domain sizes. A two-step successive DLPT consisting of BPIII-to-BPII and BPII-to-BPI as well as a three-step successive DLPT consisting of BPIII-to-BPI, BPIII-to-BPII, and BPII-to-BPI are investigated. The evolution of the lasing properties during the phase-transition process demonstrates that the thermoelastic martensitic transformation is a dual-stage process. After a crystalline grain is freshly formed, a gradual improvement in the quality of the crystal lattice is observed with the progress of the phase-transition process. Understanding the detailed DLPT mechanism of this soft cubic superstructure will provide important insights into the design and fabrication of BPLC-based functional optoelectronic materials and devices.

## Methods

**Materials**. The following nematic LC, reactive LC, left-handed chiral dopant, photoinitiator, crosslinker, and laser dye are used in this study. A commercial unreactive hybrid nematic LC (HTG135200, $n_o = 1.5143$, $n_e = 1.7136$, $n_{avg}$ = 1.5807, and $\Delta n = 0.1993$ at 20 °C; Jiangsu Hecheng Co., Ltd.) was employed as the host LCs. Diacrylate C6M (Merck) is a UV-reactive monomer. In addition, R5011 (Jiangsu Hecheng Co., Ltd.) was used to induce the chiral nematic phase owing to its high helical twisting power, and I-651 (J&K Scientific Ltd.) was used as a photoinitiator. The non-mesogenic crosslinker TMPTA (J&K Scientific Ltd.) was applied to produce the polymer network. The laser dye was 3-(2′-benzothiazolyl)-7-diethylaminocoumarin (C6) (Aldrich). All solvents and chemicals were of reagent quality and used without further purification. The chemical structures are listed in Table S1. To fabricate the LC cell, two glass substrates were cleaned sequentially using a detergent, isopropanol, ethanol, and deionized water in an ultrasonic environment for 30 min each and then dried using flowing nitrogen.

**Photopolymerization of the transition stage**. The neat green reflecting film was composed of HTG135200 (30 wt%), C6M (61 wt%), R5011(3.5 wt%), TMPTA (4.0 wt%), and I-651 (1.5 wt%). The film used in the lasing experiments was composed of HTG135200 (30 wt%), C6M (60 wt%), R5011 (3.5 wt%), TMPTA (4.0 wt%), I-651 (1 wt%), and C6 (1.5 wt%). The films were prepared as follows. First, the reaction mixture was dissolved in $CH_2Cl_2$ to form a uniform solution. Next, the solvent was evaporated through a rotary evaporation, and a uniform mixture was obtained. The mixture was heated to an isotropic state (~90 °C, ~10 °C above its clear points) and injected into a sandwiched cell through a capillary action on a heating stage. The cells used to prepare the polydomain BPLCs consisted of two glass substrates without any surface treatment, and polyethylene terephthalate (PET) films with a thickness of 50 μm were used as cell spacers. The polymerization temperature of the samples was controlled by a hot stage equipped with a programmable temperature controller. All samples were heated to an isotropic state and then gradually cooled to room temperature at 0.05 °C/min. The transition temperatures of all samples were determined using POM. When the appropriate temperature for each stage appears, the temperature is maintained at this level for 10 min to allow the stage to approach its thermodynamically stable state. Subsequently, UV radiation with an intensity of 0.45 mW/cm² and $\lambda_c$ = 365 nm was applied to the cells to cure the precursors for 30 min. I-651 absorbs UV light to induce polymerization. After UV irradiation, PS-BPLCs were formed. A number of PS-BPLC films were removed from the LC cells and cut into strips with dimensions of 2 × 20 mm to make Mobius strips.

**Fabrication of monodomain BPLCs**. Monodomain BPLCs were fabricated on clean glass substrates coated with polyimide (PI) layers. The substrates were treated by spin-coating a solution of 2.5 wt% N-methyl-2-pyrrolidone (NMP) in poly-acrylamide (PAA (BPDA/DPE)) on the glass substrates, where a solution of 10% NMP (1.0 g) in PAA (BPDA/DPE) was diluted with NMP (1.0 g) and γ-butyr-olactone (2.0 g). The treated substrates were prebaked at 100 °C for 1 h and then baked at 250 °C for 1 h to obtain a planar PI layer on the substrates. These PI layers were mechanically buffed to produce a uniform pre-tilt angle. The top and bottom glass substrates were attached to each other in an antiparallel configuration, and 50-μm thick PET films were used as cell spacers. The remaining steps for preparing monodomain BPLCs are the same as those described above for polydomain PS-BPLCs.

**Angle-resolved microspectroscopy**. The polydomain and monodomain BPLCs were measured, and the results are represented in a contour plot. The BPLC samples were observed under an optical microscope in transmission and reflection mode under illumination by a tungsten lamp. The transmitted/reflected real-space-resolved signals are transformed into momentum-space-resolved (angle-resolved) signals after passing through an objective lens, and the signals are then spatially filtered using a slit. The signals were captured using a spectrometer to obtain a high-frequency resolution (Fig. S51). An in situ ARM observation is conducted during cooling by fixing the samples on a hot stage and measuring them using the same experimental setup. The normalized frequency is calculated as $f_{nor} = \frac{\omega a}{2\pi c}$ ($\omega$, angular frequency of light; $a$, lattice constant; and $c$, light speed). For BPII and BPI, $f_{nor}$ was calculated using $a_{BPII} = 167.33$ nm and $a_{BPI} = 254.48$ nm.

**Optical characteristics**. Microscopy images of the BPLCs were taken using an inverted microscope in reflection mode. The textures of the BPLCs were observed through cross-polarized optical microscopy. The Bragg-reflected wavelengths were measured at a normal angle using an Ocean Optics spectrometer and a POM microscope with a standard reflective aluminum mirror (Shanghai Ideaoptics Corp., Ltd.) used as a reference. The temperature-dependent spectra were obtained using this system with a heating stage.

**Transmission electron microscopy**. After the cover slide is removed, portions of the sample with BPLC platelets are scratched away. The films obtained were cut into small pieces of 2 × 2 mm in size and embedded in an epoxy resin (Quetol 812 set, NisshinEM, 340). Resins with the embedded PS-BPLCs films were

maintained at room temperature. After being fully cured, the resins are cut into thin slices with a thickness of ~50 nm using an ultramicrotome (LEICA, EM UC7). The cuts were made parallel to the film plane, and the resins were deposited using a fishing method on copper grids covered with carbon films[47]. The TEM images were recorded on a Hitachi HT7700 and JEM-1011 instrument operating at 200 kV.

**Lasing behavior measurements**. The lasing actions were characterized using a Nikon micro-imaging spectrometer with a 10x objective lens. A Ti:Al$_2$O$_3$ (titanium-doped sapphire) second harmonic generation pulse laser ($\lambda_c$ = 400 nm) with a pulse duration of 180 fs and a repetition rate of 1 kHz was used as the incident laser. To detect the lasing spectra, the spectrometer was equipped with a monochromator with gratings of 300, 800, and 1200 lines and a liquid-nitrogen cooled intensified charge-coupled device detector. The maximum resolution of the spectrometer was ~0.023 nm. The samples subjected to laser spectroscopy were measured at room temperature. The excitation intensity was varied using neutral density filters. The experimental setup is shown schematically in Fig. S68. The spot size of the light source can be calculated using $d = 1.44\lambda_p/N_A$, where $\lambda_p$ represents the $\lambda_c$ of the pump laser and $N_A$ is the numerical aperture. In this experiment, $d$ was calculated to be 1.63 μm, which is significantly smaller than the size of one domain.

**Syn-SAXS**. Syn-SAXS experiments were conducted on a 1W2A SAXS beamline at the Beijing Synchrotron Radiation Facility. Syn-SAXS data were collected from free-standing PS-BPLCs at each stage. X-rays with a wavelength of 1.54 Å were used. The beam size was 1 × 2 mm, and the position of the detector was adjusted to 5 m, making it possible to cover a broad range of $q$ vectors corresponding to periodicities of ~5–300 nm. The one-dimensional syn-SAXS profiles were extracted from the measured two-dimensional syn-SAXS patterns along a specified direction using FIT2D software.

**Simulation of band structures**. The band structures of BPI and BPII along the high-symmetry directions of the irreducible BZs were simulated using the plane-wave expansion method. In the calculated model, the DTCs are placed in an isotropic medium, and the uniaxial anisotropic medium (with dielectric constants $\varepsilon_o$ and $\varepsilon_e$) is twisted from 0° to 45° along the radius of the cylinder within the DTCs. The unit cells of BPII and BPI are used as the computational region with Bloch boundary conditions imposed on their periodic boundaries. To construct photonic band diagrams, the Bloch wave vectors are taken from the high-symmetry directions of BZs[54,55] with the high-symmetry points labeled $\Gamma$, $H$, $N$, and $P$ for BPI and $R$, $X$, $S$, and $\Gamma$ for BPII. The lengths between the high-symmetry points in the diagrams correspond directly to the $k$-space distances in the BZs. The eigenfrequencies are calculated using the chosen wave vectors ($k$) and are plotted as a function of $k$ (Fig. 4a–b). Directional bandgaps appear at the boundary of the BZs.

## Data availability

The data that support the findings of this study are available from the corresponding author upon reasonable request.

## Code availability

The models used in our calculations are available from the corresponding author upon reasonable request.

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

## Acknowledgements

We are grateful for the financial support from the National Key R&D Program of China (Grant No. 2016YFA0200803, 2017YFA0204504, 2016YFB0402004, 2016YFA0301100, 2016YFA0302000, and 2018YFA0306201), NSFC (Grant No. 51873221, 52073292, 51673207, 51673208,61822504, 51873060, 91963212, 11774063 and 11727811), the Beijing Municipal Science & Technology Commission (Z181100004418012), Chinese Academy of Sciences and Dutch research project (1A111KYSB20190072), the Science and Technology Commission of Shanghai Municipality (Grants No. 17ZR1442300 and No. 17142200100) and Beijing Natural Science Foundation (2182079). We thank Prof. Guang Mo, the member of the Institute of High Energy Physics of the Chinese Academy of Sciences, for assistance with Syn-SAXS testing by the 1W2A SAXS beamline at the Beijing Synchrotron Radiation Facility.

## Author contributions

J.X.W. and Z.G.Z. conceived the original idea, J.X.W., L.S., B.G., F.J., and J.L. designed the experiment, J.L. performed the experiments and drew the scheme. L.S., Z.W.L. and B.W. conducted the theoretical simulation and measurement of the angle-resolved spectra system in the micro-region. J.L. wrote the original manuscript, J.X.W., Z.G.Z., L.S., F.J., T.I., and L.J. revised it. All authors discussed the results and commented on the manuscript.

## Competing interests

The authors declare no competing interests.
