## [Peer Review File · Nature Communications]

Reviewers' Comments:

Reviewer #1:

Remarks to the Author:

The authors attempt to apply the concept of "diffusionless phase transition" and "thermoelastic transformations" to phase transitions between blue phases of liquid crystals. Such an approach is not justified. The data presented in the manuscript do not prove that the phase transitions in blue phases are "diffusionless" since these data do not track the dynamics at the molecular scale.

The concept of DLPT has been developed to describe first-order solid-solid phase transitions in metals and alloys, in which atoms shift over small distances (typically subatomic) but do not diffuse over large distances. These shifts result in a homogeneous restructuring of the lattice and formation of a new crystal structure, see, for example, Z. Nishiyama, *Martensitic Transformation* (Academic, New York, 1978) and D. A. Porter and K. E. Easterling, *Phase Transformation in Metals and Alloys* (Chapman and Hall, London, 1992). A good example of such a transition is the martensitic transformation in steel.

Blue phases of liquid crystals are fundamentally different from metals and alloys since the molecules in them are free to move in space. These phases are positionally ordered only in the sense of molecular orientations, not in the sense of molecular positions. The structure shows a three-dimensional periodicity of molecular orientations, but not of the positions. This is why the blue phases are fluids. Since in each blue phase the molecules are free to move around in space, the concept of the "diffusionless" phase transition does not apply to them. If the authors were to prove the DLPT in the blue phases, they should have tracked the molecular positions and demonstrate that the molecules do not diffuse. Such an experiment has not been described in the present manuscript.

The paper contains many statements that are inaccurate, misleading, or simply wrong. I present only a few examples.

1. "The clear boundary between BPIII and BPII confirms that BPIII is transformed to BPII by an adiabatic DLPT process" This statement is misleading since a clear boundary between two phases is a typical feature of the first-order transition with nucleation, and the first-order transitions are not adiabatic; furthermore, it cannot be a DLPT process since the molecules are free to move.
2. "diffusionless order-to-order transitions process has the advantages of accelerating the kinetics, reducing energy dissipation, and eliminating fatigue" This statement has no physical meaning, as there is nothing in the paper that shows a reduction in energy dissipation or elimination of fatigue.
3. "Importantly, all of the phase transition processes (BPIII→BPII, BPIII→BPI, and BPII→BPI) are diffusionless and occur on the submicron scale" This statement is partially wrong and partially trivial. Phase transitions presented in the figures occur through nucleation and thus must be accompanied by diffusion of molecules. Furthermore, it is clear that all phase transitions in conventional condensed matter occur at a submicron scale since molecules and atoms are smaller than a micron.

Reviewer #3:

Remarks to the Author:

The authors have performed an impressive thorough study of phase transitions in PS-BPLC system and provided quantitative theoretical modeling and analysis of their photonic crystalline properties. The work would be a valuable contribution to our basic understanding of these properties/processes down to sub-micron scale.

I only have some general comments:

1. From what is presented, I gather that the results [experiments and analyses] apply only to polymer stabilized BPLC. In that case, is it surprising [not expected] to have diffusion-less transitions in a highly stabilized system that has a temperature range of more than 500 °C? Do most of the major findings here apply also to pristine BPLC? If not, the title and the abstract should clearly distinguish/specify the applicable BPLC systems, and not just soft cubic superstructure.
2. The authors have studied many details concerning phase transitions. Are these details new and do they lead to significant modification of our understanding of macroscopic BPLC properties which

are ultimately the ones that matter in the applications [e.g. ref. 8-13] cited by the authors?
3. Similar to 2, but on the fundamental level, do these details lead to significant modification of, or do they just confirm our basic understanding of the crystalline transformations?
Instead of embedding the answers to 2 and 3 these queries in the texts on analysis of experimental data, it would be better if the authors can put them all in some paragraphs before the conclusion.

Reviewer #4:
None

Referee 1:

Comment 1: The authors attempt to apply the concept of “diffusionless phase transition(DLPT)” and “thermoelastic transformations” to phase transitions between blue phases of liquid crystals. Such an approach is not justified.

Response: The authors would like to thank the reviewer for the critical comment. The so-called “DLPT” and “thermoelastic transformations” are typically used to refer to a few atoms in an atomic crystal¹ (or molecules in a molecular crystal) undergoing collective and highly coordinated motion² without any change in the the neighbors of atoms or molecules. Some DLPTs, especially martensitic transformations, can occur in various types of metallic, nonmetallic crystals, and ceramics. In our experiments, BPLCs are analogous to atomic crystals³⁻⁶ based on their highly ordered structures at both the nanometer (orientational molecular order) and micrometer scales (chiral arrays of DTCs). Chirality causes the molecules to self-assemble into DTCs on the micrometer scale, which are not rigid, individual entities, but rather well-defined molecular regions (Fig. 1)⁴. In BPLCs, molecule diffusion takes place constantly, and their preferred orientation depends on their position; that is, once the positions of molecules in DTCs are determined, their average orientation in DTCs is also determined. DTCs can be considered as structural units of BPIII (3D amorphous structures, Fig. 1B₁)⁷⁻⁹, BPII (crystalline SC, Fig. 1B₂)¹⁰⁻¹⁴, or BPI (crystalline BCC, Fig. 1B₃)^{10,11,13,15-16}. In this manuscript, “DLPT” and “thermoelastic transformation” indicate that DTCs in the BPLCs undergo collective, highly coordinated motion without a change in the DTCs’ neighbors.

Fig. 1 DTC structures in BPLC. (A₁) Simplified schematic illustrations of the double twist. (A₂) Top view of the double twist. (A₃) A DTC. Reproduced under the terms of Ref. 9. (B₁-B₃) Schematic illustrations of the structures of (B₁) BPIII, (B₂) BPII, and (B₃) BPI constructed from DTCs. The green rods are simplified liquid crystal (LC) molecules that self-assemble into DTC structures, which further self-assemble or entangle to form cubic or amorphous BPLCs. Reproduced under the terms of Ref. 21.

According to the results reported in our manuscript, the phase transformation in BPLCs can be considered to be a series of soft, mesoscopic diffusionless transformations (Fig. 1) based on the diffusionless reconfiguration of DTCs, which is analogous to atomic diffusionless transformations. The phase transformation between BPII and BPI is proven to be a martensitic transformation based on the following characterization: (i) A shear termed as invariant plane strain is required¹ parallel to the habit plane. The transition between BPII and BPI is the result of strain release via twinning (Figs. 2A₁-B₂). (ii) Twins in martensite may be self-accommodating and reduce energy by surface relief¹. Surface relief in BPLCs was observed in both polydomain and single-domain BPI during phase transformation from BPII (Figs. 2C₁-C₂). (iii) The martensite and parent phases maintain coherence¹. Based on the coherent lattice of BPII and BPI, a DTC simultaneously belongs to BPII and BPI, which means that DTCs do not diffuse in the process of phase transition (Figs. 3 and 4A₁-A₂; see also Figs. S24, S26, and S40 in the revised manuscript). (iv) One of the features of the microstructures is the clear crystallographic dependence of martensite formation¹. The orientation relationships are proven to be $\{110\}_{\text{BPII}}//\{211\}_{\text{BPI}}$ and approximately $\{100\}_{\text{BPII}}//\{110\}_{\text{BPI}}$ (Fig. 5).

Fig. 2 Twins (A-B) and surface reliefs (C) observed during the phase transformation. (A₁) POM image taken in reflection mode. The phase transition from BPII to BPI occurs in a LC cell composed of two bare glass substrates without any surface treatment. Cross-hatched textures, namely twin textures, were observed in the center of the BPIII/BPII/BPI core-shell configuration, where BPI₍₁₁₀₎ (the label of BPI₍₁₁₀₎ denotes that the BPI domain with (100) crystal planes parallel to the substrates) was covered by BPII₍₁₀₀₎ and then BPIII. (A₂) Magnified POM image exhibiting twin textures. (B₁) Twin textures observed in a monodomain BPI₍₁₁₀₎ transferred from monodomain BPII₍₁₀₀₎. (B₂) Schematic of the BPI₍₁₁₀₎ lattice after martensitic transformation from monodomain BPII₍₁₀₀₎. (C₁-C₂) Surface reliefs of BPI₍₁₁₀₎ after phase transformation from BPII₍₁₀₀₎ (C₁) before and (C₂) after polymerization. (C₃) Schematic of the surface reliefs.

For the DLPT of atomic crystals, individual atomic movements are less than one interatomic spacing from the start to the completion of the transformation, and the regimented manner in which atoms change position, has led to it being termed “military”¹. In our work, although the molecules

are free to move in BPLCs, it is confirmed that the building blocks of DTCs (structural units of BPI, BPII, and BPIII) do not diffuse during the phase transition process from BPIII to BPII or from BPIII to BPI (Figs. 3, 4) which is directly characterized by TEM.

Fig. 3 TEM images (A₁-E₁), corresponding fast Fourier transform (FFT) patterns (A₂-E₂), theoretically predicted FFT patterns (A₃-E₃) of BPLCs, and theoretical models for the arrangement of DTCs (A₄-E₄). (A) BPII₍₁₀₀₎, (B) BPI₍₁₁₀₎, (C) interface between BPII₍₁₀₀₎ and BPI₍₁₁₀₎, (D) interface between BPI₍₁₁₀₎ and BPIII, and (E) interface between BPII₍₁₀₀₎ and BPIII.

Fig. 4 TEM observation of the interfaces between BPIII, BPII, and BPI. (A₁-A₂) Coherent crystal lattices between the interface of BPII and BPI. DTC arrangement of the well-matched parts is highlighted by dotted white lines. (B) Interface between BPIII and BPII, in which a DTCs can simultaneously belongs to crystal lattices of BPIII and BPII. (C₁) Interface between BPIII and BPII, where curved lines highlight the DTCs that coexist in both BPII and BPIII. (C₂) Magnified TEM image of the interface of BPI/BPIII, where the dotted lines highlight the DTCs observed across the interface between BPIII and BPI.

Fig. 5 Lattice orientation relationships between BPI and BPII undergoing a thermoelastic martensitic transformation. (A₁) $\{100\}_{\text{BPII}}$ is approximately parallel to $\{110\}_{\text{BPI}}$ while there is a small angle between them. (A₂) ARM result of BPLCs captured at 77.0 °C (Stage III) exhibit a well-matched dip of the $\{110\}_{\text{BPI}}$ streak with $\{100\}_{\text{BPII}}$ streak. The arrows highlight the centers of the streaks. (A₃) ARM result of BPLCs captured at 74.8 °C (Stage IV) exhibits a slightly mismatched center of the BPI_{110} streak in relation to BPII_{100}. As the center of BPI does not shift, the mismatch may be caused by the

rotation of $\{100\}_{\text{BPII}}$, resulting in a small angle between the approximately parallel $\{100\}_{\text{BPII}}$ and $\{110\}_{\text{BPI}}$. The arrows highlight the centers of the streaks. (B₁) Schematic illustration of the orientation relationship between BPI and BPII where $\{110\}_{\text{BPII}}$ is parallel to $\{211\}_{\text{BPI}}$. (B₂) Syn-SAXS pattern measured from the BPLCs with BPI₍₁₁₀₎ and BPII₍₁₀₀₎ in coexistence, which is polymer-stabilized at 77.0 °C. The speckles of BPI₍₂₁₁₎ and BPII₍₁₁₀₎ are relatively close, indicating that $\{110\}_{\text{BPII}}$ and $\{100\}_{\text{BPI}}$ are almost parallel. (B₃) Syn-SAXS pattern measured from the BPLCs with BPI₍₁₁₀₎ and BPII₍₁₀₀₎ in coexistence, which are polymer-stabilized at 76.5 °C. The speckles of $\{211\}_{\text{BPI}}$ and $\{110\}_{\text{BPII}}$ overlap, indicating that $\{110\}_{\text{BPII}}$ and $\{100\}_{\text{BPI}}$ are parallel.

To investigate the phase transformation behavior between BPIII and BPI or BPIII and BPII, direct observation of the arrangement of DTCs is required. However, because the phase transition processes are fast and sensitive to temperature, it is difficult to track the DTCs of BPLCs with liquid-like nature. In our work, the phase transition processes between BPIII and BPI or BPIII and BPII can be separately frozen by photo-polymerization to form PS-BPLCs with well-preserved microstructures and optical properties.

DTCs with the diameter of several tens of nanometers, as structural units of BPLCs, were directly observed and proven by Yoshida's¹⁶ and Kikuchi's^{17,18} groups using confocal laser scanning microscopy and TEM. In our study, Figs. 3A₁-E₁ are the TEM images, Figs. 3A₄-E₄ are the theoretical models for the arrangement of the DTCs considering that the director of the molecule is on average oriented parallel to the cylinder axis within the DTCs, the DTCs appear bright when oriented perpendicular to the slice plane and dark when parallel to it¹⁶. Because the fast Fourier-transform (FFT) patterns transferred from the experimental results (Figs. 3A₁-E₁) match well with those transferred from the theoretically predicted model (Figs. 3A₄-E₄), the theoretical model can represent the arrangement of DTCs in Figs. 3A₁-E₁, ensuring that the arrangement of DTCs can be directly observed in real-space by TEM.

Here, although BPIII is amorphous, DTCs as structural units organized from the molecules do not diffuse during the phase transition process. The phase transformation from BPIII to BPI or BPII is diffusionless based on the rearrangement of DTCs from amorphous BPIII to BPII (with SC structures) or BPI (with BCC structures). The evidence is listed as follows: (i) DTCs are observed across the interface of BPIII/BPII (Fig. 4B) or BPIII/BPI (Figs. 4C₁-C₂). (ii) A DTC simultaneously exists in both BPII units and BPIII (Fig. 4B-C). (iii) No transition area of diffused DTC (area of molecules with random arrangement) is observed near the interfaces between BPII and BPIII or BPI (Figs. 2D-E, S23, S27, S30D-E in the revised manuscript). (iv) Thanks to the unusual phase transformation

from BPIII to BPI (Figs. S7A-D, S10F₁-F₄ in the revised manuscript), there always exists a layer of BPIII between the isotropic state and the BPI (Figs. 1C₂, S3, S7, S8, and S10 in the revised manuscript) or BPII (Figs. 1D₂, S6F, S10F-G in the revised manuscript), proving that the formation of BPII or BPI requires pre-formed DTCs of BPIII because BPII or BPI cannot directly form from the isotropic state.

As for the thermoelastic martensitic transformation, the reversible phase transformation with small thermal hysteresis and good geometric compatibility proceeds only while the temperature is changing^{2,19,20}. The phase transformation starts after the nucleation of BPI and proceeds as the temperature falls. When the cooling is stopped, the phase transformation stops, and when the cooling is resumed, it starts again. The phase transformation proceeds only while the temperature changes. Herein, the phase transformation from BPII to BPI is further proven as a thermoelastic martensitic transformation based on the following: (i) the martensitic nuclei of BPI are formed in the center of BPIII/BPII core-shell configurations (highlighted by the white circle in Fig.6 A; see also Figs. S3, S6, and S10C-11A in the revised manuscript) at a certain temperature, and the domains of BPI grow to a certain size instantaneously. Subsequently, the formed martensite BPI domain continues to grow and thicken with a decrease in temperature; that is, BPI can grow at variable temperature (Fig. S12). The textures of the BPIII/BPII/BPI core-shell configurations being held at a certain temperature can be preserved at least 24 h (Fig. S13 in the revised manuscript). (ii) The forward and backward martensitic transformations between BPII and BPI are completely reversible for more than 50 conversion-reversion cycles (see Movie S2 in the revised manuscript). (iii) The perfectly reversible phase transformation between BPI and BPII is rapid with little thermal hysteresis (Fig. 6), proving that the phase transition process is always an equilibrium process at different temperatures. (iv) The lattices of BPII and BPI are coherent with good geometric compatibility, which is proven by the similar q values corresponding to BPII_{110} and BPI_{211} (Figs. S41B₃-E₃, S42C₁-D₁, S43, S49B₂ in the revised manuscript). A DTC simultaneously belongs to a BPII and BPI unit (Figs. S24, S26 and S40A in the revised manuscript), confirming that the DTC does not diffuse in the process of phase transition. (v) Both the BPII (SC) and BPI (BCC) composed of DTCs are ordered structure, and the macroscopic volume change caused by the phase transformation are small. Herein, the transformation from BPII to BPI takes place within 2 s after a slight decrease in the temperature of the upper surface of the LC cell by flowing cold air. The reversible transformation from BPI to BPII takes place within 10 s when

the temperature of the upper surface of the LC cell recovers to the original state by stopping flowing cold air (Fig. 6). **Please see:** Figs. S4-5, S8, S10-11, S21-22, S24-25, S28, S38, S40-41, S45-47, and Movie S2 in the revised manuscript.

Fig. 6 POM observation of complete reversibility and fast phase transition process between BPI and BPII. (A-D) Textures of Stage IV when the temperature of the hot stage is maintained at 76.3 °C upon flowing or stopping flowing cold air for the same region. (A) Original state of Stage IV. (B) Phase transition occurring within 2 s from BPII to BPI after flowing cold air. (C) Phases transferring from BPI back to BPII when stopping flowing cold air after 10 s. (D) Phase transferring from BPII to BPI after flowing cold air within 2 s.

Comment 2: The data presented in the manuscript do not prove that the phase transitions in blue phases are “diffusionless” since these data do not track the dynamics at the molecular scale.

Response: The authors would like to thank the reviewer for the critical comment. As discussed in Comment 1, soft DTC structures are considered as structural units for the construction of BPIII⁷⁻⁹, BPII¹⁰⁻¹⁴, and BPI^{10,11,13,15-16}. The building blocks of DTCs are proven to be diffusionless, exhibiting collective and highly coordinated motion during the phase transformation of BPLCs. The data presented in the revised manuscript were collected by dynamically tracking the DLPT behavior of DTCs rather than the motion of molecules because the diffusion of molecules takes place constantly, and their preferred orientation in DTCs depends on their position.

In our work, the phase transition from BPIII and BPII to BPI can be divided into five stages (Figs.1, and S2-3 in the revised manuscript) during the cooling process depending on the coexistence of BPI, BPII, and BPIII. First, BPIII with amorphous structures consists of randomly distributed DTCs formed in an isotropic background, in which the DTCs are self-assembled from the LC molecules. Subsequently, the randomly distributed DTCs collectively rearrange into SC BPII or BCC BPI upon the phase transition temperature, resulting in the DPLT from BPIII to BPI or BPII. As for the phase transformation from BPII to BPI, the process is proved as a thermoelastic martensitic transformation, in which the DTCs in SC BPII reconfigure and rotate to BCC BPI.

In each stage, the microstructures and optical properties are fixed by polymer-stabilization of the liquid-like BPLCs to form PS-BPLCs, in which the arrangement of DTCs can represent the

arrangement of liquid-like BPLCs. Based on the high stability of PS-BPLCs, the BPLCs in different stages of the phase transition process are tracked by TEM (Figs.2, S23-27, S30, and S40 in the revised manuscript), POM (Figs.1A₂-E₂ and S1-14 in the revised manuscript), syn-SAXS (Figs. S41-45 in the revised manuscript), and ARM (Figs. S47-62). The characteristic results of the distribution of DTCs, the evolution of textures during the phase transition process, and dynamic tracking of the optical properties confirm that all of the phase transition processes in BPLCs (BPIII→BPII, BPIII→BPI, and BPII→BPI) are diffusionless, especially the phase transformation between BPII and BPI is a thermoelastic martensitic transformation. During the cooling process, it is found that the DTCs, which are organized from LC molecules, are randomly distributed in amorphous BPIII. Subsequently, two types of successive DLPT are confirmed when the BPLCs transfer from Stage I to V: (i) Two-step process consists of BPIII→BPII and BPII→BPI sequentially, resulting in sequential transitions from BPIII to the BPIII/BPII and BPIII/BPII/BPI core-shell configurations and finally to BPI (Figs. S10A-D in the revised manuscript). The DTCs reconfigure from amorphous distribution to SC structures, and then to BCC structures without diffusion. (ii) Three-step process consists of BPIII→BPI, BPIII→BPII, and BPII→BPI sequentially, which causes sequential transitions from BPIII domain to BPIII/BPI and BPIII/BPII/BPI core-shell configurations and finally to BPI (Figs. S10E-H). The DTCs reconfigure from amorphous distribution (BPIII) directly to BCC (BPI) structures, and then the residual amorphous distributed DTCs (BPIII) reconfigure to SC structure (BPII) without diffusion. Finally, all the DTCs with SC structure (BPII) reconfigure to BCC structure(BPI).

The TEM results show the interfaces of BPI/BPII (Figs. 2C, S24-26 S30C, and S40A in the revised manuscript), BPII/BPIII (Figs. 2E, S27, S30E, and S40 in the revised manuscript), and BPI/BPIII (Figs. 2D, S23, S30D, and S40C in the revised manuscript) during the phase transition process. In this case, the interfaces between BPI and BPII are coherent and a DTC can simultaneously belong to BPI and BPII units, confirming that the DTC undergoes a reconfiguration process without diffusion. In addition, the same DTC is observed across the interfaces of BPIII/BPI or BPIII/BPII, indicating that DTCs do not diffuse when phase transition occurs from BPIII to BPII or BPI. The interfaces of BPI/BPII, BPII/BPIII, and BPI/BPIII are clear without a nearby transitional region. The reconfiguration of DTCs occurs on a submicrometer scale that is smaller than the lattice constant (254.48 nm for BPI and 167.33 nm for BPII).

Based on the dynamic tracking of optical textures by POM, the forward and reverse martensitic

transformation between BPII and BPI is rapid and completely reversible for more than 50 conversion-reversion cycles (Fig. S8A in the revised manuscript). The reverse transformation occurs not by renucleation, but by a gradual reduction of the existing BPI domain, which appears to be a reversal of the forward transformation (Movie S2 in the revised manuscript) with little thermal hysteresis. The above-mentioned phenomena can be considered as typical characteristics of the thermoelastic behavior of the phase transformation between BPI and BPII. The surface relief (Figs. 2C₁-C₃) in BPI₍₁₁₀₎ before and after polymerization, as a characteristic property of martensitic transformation, is observed due to the invariant plane strain during the phase transformation from BPI₍₁₁₀₎ to BPII₍₁₀₀₎. In addition, it is found that a thin shell of BPIII always exists between the isotropic state and BPII or BPI. No BPII or BPI exhibits direct nuclei in the isotropic state (Figs. 1D2, S6F, and S10G in the revised manuscript), suggesting that BPII and BPI cannot transfer directly from the isotropic state without a mesophase of BPIII. This may be attributed to the formation of BPII, which undergoes a diffusionless reconfiguration of DTCs from BPIII. The isotropic state has no DTC, and thus BPI and BPII cannot directly transfer from the isotropic state in terms of dynamics.

Syn-SAXS demonstrates the reciprocal-space of the arrangement of DTCs. Both the rotation of the parent BPII phase and the orientation relationship of thermoelastic martensitic transformations can also be confirmed by the syn-SAXS results. In the early stage of phase transition (Stage III), an approximately parallel relationship between $\{211\}_{\text{BPI}}$ and $\{110\}_{\text{BPI}}$ can be observed, corroborated by the close but not overlapping speckles shown in Figs. 5B₂ and 5B₄. However, BPII rotates during the phase transformation, resulting in BPII_{110} parallels to BPI_{211} (Fig. 5B₁), as confirmed by the overlap of the speckles of BPI_{211} and BPII_{110} in a monodomain PS-BPLCs hybrid (Fig. 5B₃). In addition, the q values corresponding to BPII_{110} and BPI_{211} are similar, providing the coherence of the crystal lattice between BPII_{110} and BPI_{211}, which proves the thermoelastic martensitic transformation.

In terms of the angle-resolved microspectroscopy, it is observed that the dips of BPI and BPII streaks are well-matched in the early stage (77.0 °C in Figs. 5A₂ and 5B₂) of phase transition (Fig. 5A₂), indicating that BPI₍₁₁₀₎//BPII₍₁₀₀₎ (A₁). When the phase transition is in the last stage (74.8 °C in Fig. 5A₃ or 76.5 °C in Fig. 5B₃), a small deviation of the dips is observed (Fig. 5A₃) (highlighted by white arrow), resulting in a small angle between $\{110\}_{\text{BPI}}$ and $\{100\}_{\text{BPI}}$. It can be seen that the dip of BPI₍₁₁₀₎ does not change, whereas that of BPII₍₁₀₀₎ does, which means that the rotations of the parent

lattice are required during the thermoelastic martensitic transformation. **Please see:** Figs. 2, S21-36, and S59 in the revised manuscript.

Comment 3: The concept of DLPT has been developed to describe first-order solid-solid phase transitions in metals and alloys, in which atoms shift over small distances (typically subatomic) but do not diffuse over large distances. These shifts result in a homogeneous restructuring of the lattice and formation of a new crystal structure, see, for example, Z. Nishiyama, *Martensitic Transformation* (Academic, New York, 1978) and D. A. Porter and K. E. Easterling, *Phase Transformation in Metals and Alloys* (Chapman and Hall, London, 1992). A good example of such a transition is the martensitic transformation in steel.

Response: The authors would like to thank the reviewer for the helpful comment. The authors have carefully read the literatures on phase transformation, for example, W. H. de Jeu (auth.), S. Martellucci, A. N. Chester (eds.), *Phase Transitions in Liquid Crystals* (Springer Science+Business Media, New York, 1992), Z. Nishiyama, *Martensitic Transformation* (Academic, New York, 1978), and D. A. Porter and K. E. Easterling, *Phase Transformation in Metals and Alloys* (Chapman and Hall, London, 1992). DLPT is usually investigated in atomic crystals and is considered as the homogeneous and cooperative movement of large numbers of atoms over distances smaller than their nearest-neighbor spacing. In our work, DTCs are analogous to atoms with diameters of several tens of nanometers, which shift over a small distance within a crystal lattice rather than diffusing over large distances during phase transformation. In addition, the solid crystal–crystal transitions of atomic crystals are found to be strongly first-order, and the phase transformation of BPLCs shows little temperature hysteresis and appears simultaneously throughout the LC cell. Previous experimental results²¹ have shown that the transition of LCs involves latent heat, and it can be considered as the weak first order^{4,22}.

Comment 4: Blue phases of liquid crystals are fundamentally different from metals and alloys since the molecules in them are free to move in space. These phases are positionally ordered only in the sense of molecular orientations, not in the sense of molecular positions. The structure shows a three-dimensional periodicity of molecular orientations, but not of the positions. This is why the blue phases are fluids. Since in each blue phase the molecules are free to move around in space, the concept of

the “diffusionless” phase transition does not apply to them.

Response: The authors would like to thank the reviewer for the constructive comment. Although the molecules can be free to move in BPLCs, the building blocks of DTCs self-assembled from molecules do not diffuse during the phase transition process (Figs. 4, 5). DTCs are considered as structural units in the construction of BPIII⁷⁻⁹, BPPII¹⁰⁻¹⁴, and BPI^{10,11,13,15-16}. During the phase transition process between BPIII, BPPII, and BPI, the DTCs exhibit a diffusionless manner analogous to a classic martensitic transformation of an atomic crystal. Moreover, DTCs, as structural units of BPLCs, were directly observed and proven by Yoshida’s¹⁶ and Kikuchi’s^{17,18} groups. The DTCs and their defect line counterparts were adopted as structural units to simulate the martensitic transformation between BPPII and BPI⁴. In our manuscript, the observed structures do not show the positions of molecules but those of DTCs, especially from the TEM images, which directly show the arrangement of DTCs during phase changes (Figs. S19-38 in the revised manuscript). In this case, molecules can flow while maintaining a 3D crystalline order on the mesoscale³. The molecules can self-assemble into DTCs, and the average orientation of molecules in DTCs is determined by their position in DTCs. This work has focused on the DLPT behavior of DTCs which are analogous to the classic DLPT found in solid crystals⁴. It has been demonstrated that DLPT can also be found in 3D liquid-crystalline mesostructures.

Comment 5: If the authors were to prove the DLPT in the blue phases, they should have tracked the molecular positions and demonstrate that the molecules do not diffuse. Such an experiment has not been described in the present manuscript.

Response: The authors would like to thank the reviewer for the critical comment. In this work, DTCs are structural units in the construction of BPLCs, in which DTCs do not diffuse during the phase transition process. However, owing to the temperature-sensitive phase transition between BPI, BPPII, and BPIII (see Figs. S6-8 in the revised manuscript), the phase transformation is rather fast (see Figs. S4-5, and S10 in the revised manuscript), and it is extremely difficult to study the transformation experimentally¹. Thus, in our system, the microstructures and DTCs can be dynamically tracked by PS-BPLCs in different stages based on the well-preserved optical properties and microstructures by polymerization. Dynamic tracking of the DTC positions was carried out by TEM (Figs. 3-4; see also Figs.2, S21-40, and S64G in the revised manuscript), syn-SAXS (Figs. 2, S41-45 and S49B in the

revised manuscript), and angle-resolved microspectroscopy (Figs. 3, S46-62 in the revised manuscript).

In this work, DTCs with diameters of tens of nanometers^{16,17,18}, as structural units of BPLCs, were directly tracked as the structural units of BPI, BPII, and BPIII. Figures 3A₁-E₁ show the TEM images, and Figs. 3A₄-E₄ are the theoretical models of the DTC arrangement. Considering that the director is on average oriented parallel to the cylinder axis within a DTC, the DTCs appear bright or dark when they are oriented perpendicular or parallel to the slice plane, respectively¹⁴. Because the FFT patterns transferred from the experimental results (Figs. 3A₁-E₁) match well with those transferred from the theoretically predicted model (Figs. 3A₄-E₄), the theoretical model can represent the arrangement of DTCs in Figs. 3A₁-E₁, in which positions of DTCs can be directly observed in real space.

Furthermore, the positions of DTCs or coherent grain boundaries were investigated with the help of the PS-BPLCs in different stages during the phase transition process, in which the interfaces of BPII/BPI, BPIII/BPII, and BPIII/BPI were investigated (Figs. 3 and 4; see also Figs. S21-40 in the revised manuscript). As for the phase transformation between BPI and BPII, in which a DTC simultaneously belongs to a BPII and BPI lattice, the interfaces are coherent (Figs. 3C₁-C₄ and 4A₁-A₂). In addition, syn-SAXS results show similar q values corresponding to BPII_{110} and BPI_{211} (Figs. S42-43 and S49B₂ in the revised manuscript), which proves the thermoelastic martensitic transformation between BPII and BPI.

Owing to the clear interfaces observed by TEM between BPIII and BPII (Fig. 4B), the BPII dots with four-fold symmetry highlighted by the white dotted grid belong to BPII_{100}, whereas the curved lines and grid highlight the coexisting DTCs in the crystal lattices of both BPII_{100} and BPIII. Thus, during the phase transformation between BPIII and BPII, the DTC reconfiguration directly forms a new phase rather than an initial diffusion process followed by reorganization.

The phase transformation between BPIII and BPI (Fig. 4C) is similar to that of BPII and BPIII. Therefore, the phase transformation between BPIII and BPI is also diffusionless.

In summary, by direct TEM observation of the arrangement of DTCs near the interfaces, several forms of evidence were obtained for DLPT between BPIII, BPII, and BPI.

Please see: Figs. 2 and S28 in the revised manuscript.

Comment 6: “The clear boundary between BPIII and BPPII confirms that BPIII is transformed to BPPII by an adiabatic DLPT process” This statement is misleading since a clear boundary between two phases is a typical feature of the first-order transition with nucleation, and the first-order transitions are not adiabatic; furthermore, it cannot be a DLPT process since the molecules are free to move.

Response: The authors would like to thank the reviewer for the constructive comment. Clear boundaries can be observed at the interfaces of BPIII/BPI, BPIII/BPPII, and BPPII/BPI. Thus, the phase transformation between BPIII, BPPII, and BPI is the first-order transition^{4,22}. The related parts have been revised in the manuscript.

BPIII, BPPII, and BPI have multilevel structures. Although the LC molecules are free to move, the DTCs organized from the molecules, which are analogous to atoms of atomic crystals³⁻⁶, do not diffuse during the phase change (relevant evidence is discussed in **Comment 5**).

Comment 7: “Diffusionless order-to-order transitions process has the advantages of accelerating the kinetics, reducing energy dissipation, and eliminating fatigue” This statement has no physical meaning, as there is nothing in the paper that shows a reduction in energy dissipation or elimination of fatigue.

Response: The authors would like to thank the reviewer for the constructive comment. The properties of “reduction in energy dissipation” and “elimination of fatigue” are usually observed to be enhanced by the martensitic transformation in steel. However, these properties were not proven in our previous manuscript, and the corresponding part has been deleted in the revised manuscript.

In our work, the phase transformation of BPLCs (as a typical soft matter) is investigated on a super-resolution scale, providing a scientific basis for a deep understanding of the thermodynamics of phase transformation and self-assembly behavior of soft matter systems, and mastery of the formation and control of self-assembly structures for further applications. The DLPT of BPLCs can be divided into five stages based on the co-existence of BPIII, BPPII, and BPI during the cooling process with core-shell configurations, which can be developed for unique applications such as reversible pattern storage (Fig. 7), micro-area lasing (Fig. 8), responsive phase transformation (Fig. 9), and fabrication of BPLCs with large domain sizes (Fig.10).

Comment 8: “Importantly, all of the phase transition processes (BPIII→BPPII, BPIII→BPI, and

BPII→BPI) are diffusionless and occur on the submicron scale” This statement is partially wrong and partially trivial. Phase transitions presented in the figures occur through nucleation and thus must be accompanied by diffusion of molecules. Furthermore, all phase transitions in conventional condensed matter occur at a submicron scale since molecules and atoms are smaller than a micron.

Response: The authors would like to thank the reviewer for the critical comment. We have carefully checked the phase transformation behavior between BPIII, BPII, and BPI in our manuscript. The nucleation and diffusionless behavior of BPI, BPII, and BPIII will be discussed separately. In terms of phase transformation between BPII and BPI, the crystal lattices of BPI and BPII are coherent. A DTC simultaneously exists in BPI and BPII units, resulting in individual DTCs (~tens of nanometers) movements of less than the lattice constant (~254.48 nm for BPI). The nucleation of BPI is a heterogeneous process similar to the common martensitic transformation². The reasons for the heterogeneous nucleation mechanisms are listed as follows: (i) the nucleation of BPII preferentially appears in the center of BPI, and the surface of BPII is not a favorable position for nucleation, as the transformation may begin with defects inside the crystals; (ii) the average number of nuclei is independent of grain size, and increases rapidly with the increase in supercooling prior to transformation; (iii) BPI nucleates first in some domains and then in others (see phase transition behavior in poly-domain BPLCs, in which the domains of BPI are separated from each other) (see Figs. 1A₂-E₂, 3A, S1-3, S6-8, and S10-11 in the revised manuscript).

As for the transformation from BPIII to BPII or from BPIII to BPI, the DTCs undergo collective motion during the phase transformation process. The structure of BPIII has a multilevel structure, in which the BPIII is amorphous, consisting of DTCs (Fig. 1B₁), while the DTCs are self-assembled by LC molecules (Fig. 1A). Although molecules and atoms can flow in the system, the DTCs self-assembled from molecules cannot diffuse during the phase transition, resulting in the diffusionless motion of DTCs in phase transformation from BPIII to BPI or from BPIII to BPII (Fig. 4B-C). In addition, the phase transition process is fast in soft matter (Figs. S5-10F)

The typical lattice constants of BPI and BPII are 254.48 and 167.33 nm, respectively, as measured by syn-SAXS. DTCs with diameters of several tens of nanometers are regarded as assembling structural units analogous to atoms in atomic crystals, in which the movement of DTCs during the phase transformation process is less than one lattice constant. **Please see:** Figs. 2, S30, S40, S42-43, and Movie 3 in the revised manuscript.

Reference:

- 1 Porter DA, Easterling KE, Sherif M Phase Transformations in Metals and Alloys (CRC Press, Boca Raton, FL) (1992)
- 2 Olson, G. Martensitic transformation. (Academic, New York) (1980).
- 3 Li, X. *et al.* Sculpted grain boundaries in soft crystals. *Sci Adv* **5**, eaax9112-eaax9121, doi:10.1126/sciadv.aax9112 (2019).
- 4 Li, X. *et al.* Mesoscale martensitic transformation in single crystals of topological defects. *Proc. Natl. Acad. Sci. U S A* **114**, 10011-10016, doi:10.1073/pnas.1711207114 (2017).
- 5 Olson, G. B. & Hartman, H. Martensite and Life : Displacive Transformations as Biological Processes. *Le Journal de Physique Colloques* **43**, C4-855-C854-865, doi:10.1051/jphyscol:19824140 (1982).
- 6 Chen, C. W. *et al.* Large three-dimensional photonic crystals based on monocrystalline liquid crystal blue phases. *Nat. Commun.* **8**, 727-735, doi:10.1038/s41467-017-00822-y (2017).
- 7 Zasadzinski, J. A. N., Meiboom, S., Sammon, M. J. & Berreman, D. W. Freeze-Fracture Electron-Microscope Observations of the Blue Phase III. *Phys. Rev. Lett.* **57**, 364-367 (1986).
- 8 Gandhi, S. S. & Chien, L. C. Unraveling the Mystery of the Blue Fog: Structure, Properties, and Applications of Amorphous Blue Phase III. *Adv. Mater.* **29**, 1704296-1704309, doi:10.1002/adma.201704296 (2017).
- 9 Gandhi, S. S., Kim, M. S., Hwang, J. Y. & Chien, L. C. Electro-optical Memory of a Nanoengineered Amorphous Blue-Phase-III Polymer Scaffold. *Adv. Mater.* **28**, 8998-9005, doi:10.1002/adma.201603226 (2016).
- 10 Ravnik, M., Alexander, G. P., Yeomans, J. M. & Zumer, S. Three-dimensional colloidal crystals in liquid crystalline blue phases. *Proc Natl Acad Sci U S A* **108**, 5188-5192, doi:10.1073/pnas.1015831108 (2011).
- 11 Guo, D. Y. *et al.* Reconfiguration of three-dimensional liquid-crystalline photonic crystals by electrostriction. *Nat. Mater.* **19**, 94-101, doi:10.1038/s41563-019-0512-3 (2020).
- 12 Morris, H. C. S. Liquid-crystal lasers. *Nature Photonics* **4**, 676-685, doi:10.1038/nphoton.2010.184 (2010).
- 13 Salamonczyk, M. *et al.* Structure of nanoscale-pitch helical phases: blue phase and twist-bend nematic phase resolved by resonant soft X-ray scattering. *Soft Matter* **13**, 6694-6699, doi:10.1039/c7sm00967d (2017).
- 14 Gharbi, M. A. *et al.* Reversible Nanoparticle Cubic Lattices in Blue Phase Liquid Crystals. *ACS Nano* **10**, 3410-3415, doi:10.1021/acsnano.5b07379 (2016).
- 15 Kikuchi, H., Yokota, M., Hisakado, Y., Yang, H. & Kajiyama, T. Polymer-stabilized liquid crystal blue phases. *Nat Mater* **1**, 64-68, doi:10.1038/nmat712 (2002).
- 16 Tanaka, S. *et al.* Double-twist cylinders in liquid crystalline cholesteric blue phases observed by transmission electron microscopy. *Sci Rep* **5**, 16180-16189, doi:10.1038/srep16180 (2015).
- 17 Yoshizawa, D., Okumura, Y., Yamamoto, J. & Kikuchi, H. Decreasing the operating voltage of a polymer-stabilized blue phase based on intermolecular affinity. *Polymer Journal* **51**, 667-673, doi:10.1038/s41428-019-0183-6 (2019).
- 18 Higashiguchi, K., Yasui, K. & Kikuchi, H. Direct observation of polymer-stabilized blue phase I structure with confocal laser scanning microscope. *J Am Chem Soc* **130**, 6326-6327, doi:10.1021/ja801553g (2008).
- 19 Cui, J. *et al.* Combinatorial search of thermoelastic shape-memory alloys with extremely small hysteresis width. *Nature Materials* **5**, 286-290, doi:10.1038/nmat1593 (2006).
- 20 Kaushik, B., Sergio, C., Giovanni, Z. & Johannes, Z. Crystal symmetry and the reversibility of martensitic transformations. *Nature* **428**, 55-59, doi:10.1038/nature02378 (2004).

- 21 Stegemeyer, H., Blümel, T. H., Hiltrop, K., Onusseit, H. & Porsch, F. Thermodynamic, structural and morphological studies on liquid-crystalline blue phases. *Liquid Crystals* **1**, 3-28, doi:10.1080/02678298608086486 (1987).
- 22 Thoen, J. Calorimetric studies of liquid crystal phase transitions: steady state adiabatic techniques. eds Chester AN, Martellucci S (Springer Science+Business Media, New York) (1992).

Referee 2:

Comment 1: The authors have performed an impressive thorough study of phase transitions in the PS-BPLCs system and provided quantitative theoretical modeling and analysis of their photonic crystalline properties. The work would be a valuable contribution to our basic understanding of these properties/processes down to the sub-micron scale.

Response: Authors would like to thank the reviewer for the positive evaluation.

Comment 2: From what is presented, I gather that the results [experiments and analyses] apply only to polymer-stabilized BPLCs. In that case, is it surprising [not expected] to have diffusion-less transitions in a highly stabilized system that has a temperature range of more than 500 °C? Do most of the major findings here apply also to pristine BPLC? If not, the title and the abstract should clearly distinguish/specify the applicable BPLCs systems, and not just the soft cubic superstructure.

Response: The authors would like to thank the reviewer for the constructive comment; however, there has been some misunderstanding between the authors and the referees. Herein, the authors would like to attempt to describe this as clearly as possible. The properties of DLPT were investigated in a liquid-like system, in which the BPLCs were not polymerized, and the main results are available for the pristine BPLCs. In this case, the molecules and DTCs can move, and the 3D crystalline order can reconfigure collectively. Some *in situ* tracking of the transformations of BPIII→BPII, BPII→BPI, and BPIII→BPI was conducted for liquid-like BPLCs before polymerization, as shown in Figs. 1A₂-E₂, 3A, S1-12, and S47-48 in the revised manuscript.

On the other hand, owing to the perfect preservation of microstructures and optical properties of PS-BPLCs, the phase transformation behavior of BPLCs can be tracked with the help of PS-BPLCs in different stages during the cooling process. When the liquid-like BPLCs are polymer-stabilized to PS-BPLCs, no obvious changes, including phase transformation, can be observed within an ultrawide temperature range of more than 500 °C (from -190 °C to 340 °C). In this case, the microstructures of BPLCs represent those of pristine BPLCs. Thus, some characterizations consisting of TEM, syn-SAXS, and ARM are carried out to investigate the detailed microstructures of BPLCs. **Please see:** Figs. 1A₂-E₂, 3A, S1-12, and S47-48 in the revised manuscript.

Comment 3: The authors have studied many details concerning phase transitions. Are these details new and do they lead to significant modification of our understanding of macroscopic BPLC properties which are ultimately the ones that matter in the applications [e.g. ref. 8-13] cited by the authors?

Response: The authors would like to thank the reviewer for the helpful comment. The DLPT of BPLCs can be divided into five stages based on the co-existence of BPIII, BPII, and BPI during the cooling process with core-shell configurations, which can be developed for some unique applications such as temperature-switchable binary and ternary quick response (QR) codes (Fig. 7), micro-area lasing (Fig. 8), responsive phase transformation (Fig. 9), and fabrication of BPLCs with large domain sizes (Fig.10). The details of these are as follows:

(1) Reversible pattern storage (Fig. 7)

The random distributed texture of BPLC is unique, which can be applied as the binary and ternary QR codes as shown in Fig. 7. Here, the binary (Fig. 7A₄) and ternary (Fig. 7A₅) QR codes (with large size of 37×37 pixels) are constructed from POM image (Fig. 7A₁) experienced the processes of gray-level conversion (Fig. 7A₂ and B₂), contrast change (Fig. 7A₃ and B₃), the pixelation (Fig. 7A₄ and B₄), and the encoding. The conversion is carried out mainly based on the brightness of POM image, and each domain in (Fig. 7A₁) can be regarded as a pixel of the QR code. For example, binary code can be obtained by encoding the pixels of “white” or “black” into “1” or “0”, the codes can be further related to the other codes (such as ASCII binary codes). Similarly, a ternary QR code (Fig. 7A₅-B₅) is constructed based on the random distribution of black, blue, and green colors in the POM image (Fig.7A₁) owing to randomly distribution of BPIII/BPII/BPI core-shell configurations in the black isotropic background. The ternary QR code contains a high-density information: encoding the green pixels originated from BPI₍₁₁₀₎, blue pixels from BPII₍₁₀₀₎ and the black pixels from the isotropic background into “2”, “1”, and “0” (Fig. 7B₅) respectively. The DLPT process of BPLC produces a temperature-switchable QR codes (Fig. 7). Followed by heating the hybrid LCs to the isotropic state (Fig. 7C), the sample are cooled to Stage III (76.7 °C) at 0.05 °C/min, forming a texture consisting of the random distributed BPIII domain, BPIII/BPII, BPIII/BPI, and BPIII/BPII/BPI configurations (Fig. 7D₁) and contributing to a QR code (Fig. 7D₂). When decreasing the temperature of the sample to 76.3 °C, a texture consisting of BPIII/BPII/BPI core-shell configurations are formed and contribute to a new QR code (Fig. 7E₂). Notably, the QR code in Fig. 7E₂ cannot recover to that in Fig. 7D₂ by increasing the temperature Since the phase transformation of BPIII→BPII is not completely reversible with a large thermal hysteresis. In contrast, a temperature-reversible switching of textures (QR codes) can be obtained between high- and low-temperature modes

(Fig. 7E) owing to the thermoelastic martensitic transformation between BPII and BPI. This temperature-switchable textures in soft BPLCs can be considered as an analogy to the shape-memory in solid-solid crystal transformation. In detailed, the BPI domains grow fast and lead to an increase of brightness and change of colors of the POM image (Fig. 7E₁') during the cooling process from 76.3 °C to 75.8 °C. In comparison, a completely reverse transformation from BPI to BPII occurs by gradual reduction of the existing BPI domain when heating the sample from 75.8 °C to 76.3 °C. Finally, almost all BPI are converted into BPII and the distribution of core-shell configurations is recovered to the original high-temperature mode, forming a QR code same with the initial state (Fig. 7E₁). This reversible temperature-induced phase transformation contributes to a pair of switched QR codes with high duration stability (Movie S2). Interestingly, the QR code can be reset by heating the hybrid LCs to the isotropic state (Fig. 7F), the process empty all of the texture. A pair of new temperature-switchable QR codes can be obtained by cooling the hybrid LCs to Stage IV (76.3 °C) for a second cooling process (Fig. 7G). The textures distribution and the corresponding QR codes (Fig. 7G) are distinct from those of the first cooling process (Fig. 7E), ensuring the uniqueness of the QR codes. The QR code can be further polymer-stabilized to improve thermal stability and used in a wide temperature range of from -190 to 340 °C for specific scenarios (Fig. 7H).

Fig. 7 Anti-faking ID codes based on binary and ternary temperature-switchable QR codes. (A₁) POM image of PS-BPLCs at Stage IV with low magnification. (A₂) The image obtained after the gray-level conversion of (A₁). (A₃) The image of the binarization of (A₂). (A₄) The binary codes obtained from (A₃) after pixelation and encoding processes. (A₅) Large-scale ternary QR code with the size of 37×37 pixels encodes from (A₁) undergoing ternary and pixelation processes. (B₁-B₃) The magnified image of orange squared area in (A₁-A₃). (B₄-B₅) The binary and ternary codes of the orange squared area in (A₄, A₅). In binary codes, the “black” or “white” pixel represents “0” or “1”. In ternary codes, the “green”, “blue” or “black” pixels in (A₅, B₅) represent “2”, “1” or “0”. (C) The isotropic state. (D₁) The random distributed BPIII domain, BPIII/BPII, BPIII/BPI, and BPIII/BPII/BPI configurations in Stage III obtained at 76.7 °C. (D₂) The corresponding binary QR code transferred from (D₁). The BPIII/BPII/BPI core-shell configurations in Stage IV obtained at (E₁) 76.3 °C and (E₁') 75.8 °C in the first cooling process. (E₂, E₂') The corresponding binary QR codes transferred from (E₁, E₁'). (F) The isotropic state in the second cooling process. (G₁, G₁') The BPIII/BPII/BPI core-shell configurations in Stage IV observed at (G₁) 76.3 °C and (G₁') 75.8 °C. (G₂,

G_2') The corresponding binary QR codes transferred from (G_1, G_1'). (H_1) The BPIII/BPII/BPI core-shell configurations with high thermal stability by polymer-stabilizing the textures in low-temperature mode (G_2). (H_2 - H_3) The binary and ternary QR codes transferred from the POM image (H_1) with the same methods shown in (A_1 - A_5).

(2) Micro-area lasing (Fig. 8)

Lasing has been achieved in a BPI₍₁₁₀₎ domain with a size smaller than 60 μm (Fig.8A), which means that the as-formed crystalline grain transferred diffusionlessly from BPII has a high degree of order to support the lasing. In this case, the thermoelastic martensitic transformation between BPII and BPI is fundamental for the achievement of a distributed feedback surface-emitting laser in PS-BPLCs. The crystal lattices near the phase interfaces are coherent without the transition area, which provides the possibility of high integration. The results in Fig. 8 can predict the properties of microlasers, which is promising for next-generation optical lasing devices.

Fig. 8 Bandedge lasing of BPI with domain size of approximately 60 μm . (A) Platelets structure captured by an inverted optical microscope in reflection mode; red dotted circles mark the greenish lasing phase platelets. (B) Emission spectra plotted relative to the low-energy photonic bandedge. The insets show magnified images of the lasing peaks. The measured FWHM is 0.579 nm. (C) Best fits of typical input-output curves for the data below and above the excitation threshold.

(3) Responsive phase transformation (Fig. 9)

Responsivity is also an unignorable property in next-generation optical devices¹⁻⁶. However, the properties of phase transformation from BPIII to BPII or from BPIII to BPI are still unclear due to the narrow temperature range and small domain size of BPIII. Our findings may build an understanding of the disorder-order phase transformation behavior in BPLCs.

It is found that a thin shell of BPIII exists between the isotropic state and BPI or BPII (Figs. 9A-B). No direct nuclei of BPI or BPII are observed in the isotropic state, which confirms that BPI and BPII cannot be directly transferred from an isotropic state without the formation of DTCs. It can be inferred that the formation of BPI and BPII undergoes a diffusionless reconfiguration of the DTCs of BPIII (which is further confirmed by direct observation of DTCs by TEM).

Fig. 9 POM characterization of the thin shell of BPIII between the isotropic state and BPI or BPII. (A₁-A₃) Phase transition sequentially from BPIII to BPII and then from BPII to BPI. (B₁-B₃) Phase transition sequentially from BPIII to BPI and then from BPIII to BPII.

The results reveal the reason why almost no studies have successfully achieved a stimulus response between BPII or BPI and isotropic states. This may be attributed to the fact that BPII and BPI do not exhibit direct nuclei in isotropic states without BPIII, and the formation of BPII and BPI undergoes two steps: from isotropic states to BPIII and then from BPIII to BPII or BPI. Although the phase transition process between BPIII and BPII or BPI is diffusionless, the formation of BPIII from the isotropic state is a diffusion process. Thus, the stimulating responsive phase transition between isotropic state and BPI or BPII is difficult to achieve.

In addition, the most popular responsive phase transition (responsive microstructure change) of

BPLCs is achieved in the thermoelastic martensitic transformation between BPI and BPII based on the reconfiguration of DTCs (order-order phase transformation). Here, DLPT can be considered a significant condition for stimulating responsive phase transition.

Based on our findings, the responsive disorder-order structure transformation in BPLCs can be predicted in the phase transition between BPIII and BPI or BPIII and BPII. For example, the responsive phase transformation between BPIII and BPII may be achieved by the simple addition of photoresponsive isomerization molecules. **Please see:** Figs. S3, S4, S6, S7, and S10 in the revised manuscript.

(4) Fabrication of BPLCs with large domain sizes (Figs. 10, 11)

It is observed for the first time that, the numbers of nuclei in both BPI and BPII are gradually reduced (Figs. 10A-H) after annealing for 24 h at 76.2 °C. As provides the possibility to enlarge the domain size of BPLCs, and is considered as a significant issue for improving the quality of devices fabricated based on BPLCs.

Fig. 10 POM images of BPLCs in Stage IV annealed with increasing time. (A) Optical textures of fresh-formed BPLCs by direct cooling from the isotropic state at 0.05 °C/min. (B-H) Optical textures with the temperature maintained at 76.2 °C after being preserved for 1-24 h. (I) BPI cooling from (H) at a cooling rate of 0.05 °C/min to 75 °C.

It is found that the ultra-large domain size of polydomain BPI (Fig. 11) can be enlarged based on

the reduction in the number of nuclei (Fig. 10). The reduction of nuclei of BPI and BPII after annealing for 24 h at 76.2 °C causes enlargement of the domain sizes corresponding to BPI and BPII. BPI with a large domain size over 1 mm (Figs. 11D-E), nearly three times larger than that of the sample without annealing (Figs. 11A-B), is obtained after slowly cooling at 0.05 °C/min.

Fig. 11 Enlargement of domain size by annealing. (A)(C) BPLCs in Stage III (a mixture of BPI and BPIII) cooled from the isotropic phase to 76.2 °C. (B) BPI obtained by further cooling the sample from 76.2 °C (A) to 75 °C at 0.05 °C/min. (D) Texture mixed with BPI and BPII obtained after maintaining the temperature for 24 h. (E) BPI with domain-size-enlarged BPLCs obtained by cooling from (D) at 0.05 °C/min.

Comment 4: Similar to 2, but on the fundamental level, do these details lead to significant modification of, or do they just confirm our basic understanding of the crystalline transformations?

Response: The authors would like to thank the reviewer for the constructive comment. To the best of our knowledge, our results do lead to some modifications at the understanding of the crystalline transformations at the fundamental level. In this work, we study a liquid analogy of a crystal–crystal transformation of BPLCs that exhibit long-range order and crystalline symmetries with lattice constants in the submicrometer regime. These results may be used to explain the phase transition behavior of atomic crystals and provide a possible route for its research. It is significant to the understanding of the diffusionless behavior of atomic crystals with a fast transition process that is difficult to observe directly.

Herein, owing to the evolution of lasing performance in BPI, it is found for the first time that the

thermoelastic DLPT appears to be an unprecedented dual-stage process, which is generally difficult to characterize in metal or ceramic materials owing to their complex characterization conditions. This was confirmed by lasing the evolving of ultra-narrow linewidth (or Q-factor) in different stages. Because a resonator with higher ordered structures leads to a narrower linewidth (high Q-factor) of lasing, the degree of the order corresponding to BPLCs can be justified by the linewidth (or Q-factor). In this work, the Q-factor dramatically increases from 914 to 3090 to 5054, corresponding to Stages III, IV, and V, respectively (see Figs. 1C-E in the revised manuscript). The quality of the crystal lattice is improved gradually with the growth of the BPI domain during the phase transformation. In this process, the fresh-formed crystalline grains exhibit a high degree of order, and the quality of the crystal lattice can be further improved by the growth of martensitic grains. The results are beneficial to understanding the mechanism of thermoelastic martensitic transformation through a liquid analogy of crystal–crystal transformation. **Please see:** Figs. 4 and S64-65 in the revised manuscript.

Comment 5: Instead of embedding the answers to 2 and 3 these queries in the texts on analysis of experimental data, it would be better if the authors can put them all in some paragraphs before the conclusion.

Response: The authors would like to thank the reviewer for the constructive comment. We have added a discussion regarding comments 2 and 3 to the revised manuscript before the conclusion.

Reference:

- 1 Lin, T. H. *et al.* Red, green and blue reflections enabled in an optically tunable self-organized 3D cubic nanostructured thin film. *Adv Mater* **25**, 5050-5054, doi:10.1002/adma.201300798 (2013).
- 2 Zheng, Z. G. *et al.* Light-Patterned Crystallographic Direction of a Self-Organized 3D Soft Photonic Crystal. *Adv. Mater.* **29**, 1703165-1703173, doi:10.1002/adma.201703165 (2017).
- 3 Jau, H. C., Lin, Y. T., Li, C. C., Chen, C. W. & Lin, T. H. Optically rewritable dynamic phase grating based on blue-phase-templated azobenzene liquid crystal. *Opt Express* **27**, 10580-10585, doi:10.1364/OE.27.010580 (2019).
- 4 Wang, M. *et al.* Reversible light-directed self-organized 3D liquid crystalline photonic nanostructures doped with azobenzene-functionalized bent-shaped molecules. *J. Mater. Chem. C* **6**, 7740-7744, doi:10.1039/c8tc02200c (2018).
- 5 Zhou, K. *et al.* Light-Driven Reversible Transformation between Self-Organized Simple Cubic Lattice and Helical Superstructure Enabled by a Molecular Switch Functionalized Nanocage. *Adv. Mater.* **30**, e1800237-e1800244, doi:10.1002/adma.201800237 (2018).
- 6 Kim, D.-Y. *et al.* Remote-Controllable Molecular Knob in the Mesomorphic Helical Superstructures. *Advanced Functional Materials* **26**, 4242-4251, doi:10.1002/adfm.201505342 (2016).

Reviewers' Comments:

Reviewer #1:

Remarks to the Author:

The authors' replies to my comments clarify the essence of the paper. I do not have any scientific objection and would recommend the publication provided the manuscript states clearly and from the very beginning in what sense the word "diffusionless" is used. In particular, Abstract's first claim of what is achieved, "In this study, the diffusionless phase transformation of BPLCs was comprehensively characterized using..." would be much better to restate as "In this work, we demonstrate that the structural units of the BPLC, the double-twisted cylinders (DTCs) of molecular orientation, exhibit diffusionless behavior with collective and highly coordinated motion during the phase transformations of BPLCs. The dynamics of DTCs in phase transformations was characterized comprehensively using...". The end of the Abstract, which mentions DTCs, could be slightly modified to avoid a duplicate definition of DTCs.

Similar clarifications are needed in the Conclusion, e.g., instead of "In particular, BPII->BOI is a reversible thermoelastic martensitic transformation," could be rephrased as "In particular, DTCs in BPII->BOI show a diffusionless..., which is a hallmark of a reversible thermoelastic martensitic transformation"; similar clarifications can be added for other statements in the Conclusion.

TOC entry: better to modify to "diffusionless behavior of double-twisted cylinders of molecular orientation in phase transitions..."

Reviewer #2:

Remarks to the Author:

The authors have addressed all of my comments and revised/edited the resubmitted manuscript in depth. I believe they have done an excellent and thorough study. I recommend publication of the manuscript in Nat. Commun.

Reviewer #3:

Remarks to the Author:

The authors Jie Liu et al. present in their manuscript "Diffusionless Transformation of Soft Cubic Superstructure from Amorphous, Simple Cubic to Body-Centered Cubic Phases" results on the phase transition behaviour of blue phases (BPs) in chiral nematic liquid crystals. The manuscript contains an impressive body of work that has been carried out to a high standard and provides insights at an unprecedented level of detail. It makes use of concepts and terminology of a related publication of Xiao Li et al. (PNAS 114, 10011–10016 (2017)), which confirms similar findings, mainly theoretically, but in full agreement with experiments, albeit these were carried out in a completely different experimental system. Some of this terminology may trigger wrong associations and assumptions. That said, the revised version the manuscript clears up most of these misconceptions. I recommend publication in Nature Communications provided the points below have been sufficiently addressed.

It is interesting that this martensitic transformation in the DTC structure is possible as the network of disclination lines, which is so-to-speak the counter piece to the DTCs, is topologically different in BPI and BPII. In particular, the disclination lines in BPI do not touch, while in BPII they intersect in four-way junctions and form two independent sets of interpenetrating defect networks, similar to the two opposite labyrinths of passage in gyroid phases. Hence, it is not possible to preserve the topology of the disclination network while transitioning from BPI to BPII. But a martensitic transformation of the DTCs through small local rearrangements seems to imply exactly the opposite, i.e. that there is a certain degree of topological similarity between BPI and BPII, as well as BPIII.

However, Xiao Li et al. reported that at the BPII-BPI transition a spontaneous and sudden reconfiguration takes place where the disclination lines disappear, reappear, fuse and merge to produce a new crystalline structure. A possible explanation for this striking behaviour, which was found in simulations and is also consistent with Li's experimental observations, could be that

around disclination lines the liquid crystal can be thought of as existing in its high temperature phase, therefore offering virtually no resistance to reshaping and morphing. In contrast, the DTCs are the energetically most stable conformations. It is therefore plausible that the DTC structure remains largely intact while it undergoes a collective reorganisation over larger sample areas. This could happen in a manner of recruiting new molecules to become part of DTCs and expelling others from DTCs, simply through a change in their average orientation with respect to their neighbours and akin to a travelling wave that transports only energy, but displaces no mass.

My main concern is that the experimental system is not sufficiently distinctly described as different from a pure blue phase. It consists of almost two thirds (65% wt) of polymer and cross-linker. From a practical point of view, this is of course necessary for subjecting the same samples to further experimental analysing techniques that are required. It is, however, really unclear how the dominating presence of polymer affects the phase transition behaviour. I find it problematic to apply the results directly to support statements about the general phase transition behaviour of pure blue phases, even though the individual phases are identified as BPs and the results seem to agree well with the previous results of Xiao Li et al., who studied pure, mono-crystalline blue phases that were grown using chemically patterned surfaces. I feel that this feature may be also responsible for some of the observed peculiarities like the coherence of phases across the interface separating two blue phases. If my back-of-the-envelope calculation is correct, the latent heat involved in these coherent structures is on the order of 10^6 kB T or even more and therefore very large. The authors should please emphasise why they think that despite the different experimental system these results also apply to pure BPs.

To call the transition diffusionless triggers simply the wrong associations, but follows the terminology of martensitic transformations and Xiao Li et al. As in any liquid matter, there is of course diffusion present at the molecular level. But the claim is rather that the emerging, mesoscopic, double-twist cylinder (DTC) structures do not shape and reform as a whole through diffusion, but through large and collective reorganisation. This distinction, and the presence of diffusion at the molecular level, should be made as clear as possible to avoid confusion and unnecessary criticism.

A minor point: On p.7 the authors say the transition between BPI and BPII is reversible without hysteresis. As a weak first order transition, which the pure BPI-BPII transition undoubtedly is, I think there has to be hysteresis. Moreover, the authors say in Fig. S8 that they flow cold air over the sample to trigger the transition. This sounds like a rather uncontrolled modulation of temperature, which may obscure the hysteretic behaviour. It could also be that this is somehow caused by the high level of polymer present. I think this part should be accordingly reworded.

Dear referees:

Thank you very much for the comments of our manuscript titled “Diffusionless Transformation of Soft Cubic Superstructure from Amorphous, Simple Cubic to Body-Centered Cubic Phases”(manuscript No. NCOMMS-20-09527A-Z) by Jie Liu, Wenzhe Liu, Bo Guan, Bo Wang, Lei Shi, Feng Jin, Zhigang Zheng*, Jingxia Wang*, Tomiki Ikeda and Lei Jiang. We appreciate your encouragement and the constructive evaluations. We have carefully read the comments and added supplementary experimental data as well as detailed explanations in the revised manuscript to clarify the work more understandable accordingly (the revised section is highlighted in blue, and the response to the reviewer’s comments is attached below).

A list of main changes in the revised manuscript:

Page 2, lines 36, 38-46;

Page 3, lines 67-68;

Page 4, lines 77-79, 81-82, 87, 90, 91, 95, 97-98;

Page 5, lines 115-117, 120-121, 124;

Page 6, lines 125, 127-129, 132-136, 142-145, 147-149;

Page 7, lines 151-154, 159-161, 163-165;

Page 9, lines 188, 189, 192, 204-205;

Page 10, lines 208-209, 215-216, 217-221, 227-228, 230;

Page 11, lines 242-243;

Page 14, lines 297-298;

Page 15, lines 301-304, 307, 309;

Page 18, lines 364-365, 372;

Page 19, lines 380-381, 383, 395-396, 400-402;

Page 20, lines 403, 420-421;

Page 21, lines 432, 436-438, 441;

Page 22, lines 452, 458;

Page 24, lines 476, 477;

Page 25, lines 503-506;

Page 26, lines 527-528;

Page 27, lines 529-536, 538-539;

Page 39, lines 826-827.

Supporting information:

Added Figs. S9, S16, S33, S52, S53, S54

We would appreciate if the revised manuscript could be recommended publication in *Nature Communications*. Thank you very much for your kind consideration.

Yours sincerely,
Jingxia Wang

Response to the reviewers:

Referee 1:

Comment 1: The authors' replies to my comments clarify the essence of the paper. I do not have any scientific objection and would recommend the publication provided the manuscript states clearly and from the very beginning in what sense the word “diffusionless” is used. In particular, Abstract’s first claim of what is achieved, “In this study, the diffusionless phase transformation of BPLCs was comprehensively characterized using...” would be much better to restate as “In this work, we demonstrate that the structural units of the BPLC, the double-twisted cylinders (DTCs) of molecular orientation, exhibit diffusionless behavior with collective and highly coordinated motion during the phase transformations of BPLCs. The dynamics of DTCs in phase transformations was characterized comprehensively using...”.

Response: The authors thank the reviewer for the helpful comment. We have revised the manuscript according to the comments of the reviewer. Please see: page 2, lines 38-42.

Comment 2: The end of the Abstract, which mentions DTCs, could be slightly modified to avoid a duplicate definition of DTCs.

Response: The authors thank the reviewer for the helpful comment. We have revised the manuscript according to the comments of the reviewer. Please see: page 2, lines 44-46.

Comment 3: Similar clarifications are needed in the Conclusion, e.g., instead of “In particular, BPII->BPI is a reversible thermoelastic martensitic transformation,” could be rephrased as “In particular, DTCs in BPII->BPI show a diffusionless..., which is a hallmark of a reversible thermoelastic martensitic transformation”; similar clarifications can be added for other statements in the Conclusion.

Response: The authors thank the reviewer for the helpful comment. In the conclusion, we revise “In particular, BPII->BPI is a reversible thermoelastic martensitic transformation” to “In particular, DTCs in BPII→BPI show a diffusionless, collective and highly coordinated motion, which is a hallmark of a reversible thermoelastic martensitic transformation. Besides, diffusionless behaviors of DTCs are

also proved in BPIII→BPI, BPIII→BPPII.” **Please see:** page 27, lines 529-531.

Comment 4: TOC entry: better to modify to “diffusionless behavior of double-twisted cylinders of molecular orientation in phase transitions...”

Response: The authors thank the reviewer for the helpful comment. The TOC entry has been revised to that “This paper presents a comprehensive characterization of the diffusionless behavior of double-twisted cylinders of molecular orientation in phase transition of the soft cubic superstructures...”.

Please see: page 39, paragraph 1, lines 826-827.

Referee 2:

Comment 1: The authors have addressed all of my comments and revised/edited the resubmitted manuscript in depth. I believe they have done an excellent and thorough study. I recommend publication of the manuscript in Nat. Commun.

Response: The author would like to thank the reviewer for the positive evaluation.

Referee 3:

Comment 1: The authors Jie Liu et al. present in their manuscript "Diffusionless Transformation of Soft Cubic Superstructure from Amorphous, Simple Cubic to Body-Centered Cubic Phases" results on the phase transition behavior of blue phases (BPs) in chiral nematic liquid crystals. The manuscript contains an impressive body of work that has been carried out to a high standard and provides insights at an unprecedented level of detail.

Response: The authors would like to thank the reviewer for the positive evaluation of the manuscript.

Comment 2: It makes use of concepts and terminology of a related publication of Xiao Li et al. (PNAS 114, 10011–10016 (2017)), which confirms similar findings, mainly theoretically, but in full agreement with experiments, albeit these were carried out in a completely different experimental system. Some of this terminology may trigger wrong associations and assumptions. That said, the revised version the manuscript clears up most of these misconceptions. I recommend publication in Nature Communications provided the points below have been sufficiently addressed.

Response: Authors would like to thank the reviewer for the positive evaluation of the manuscript.

Comment 3: It is interesting that this martensitic transformation in the DTC structure is possible as the network of disclination lines, which is so-to-speak the counter piece to the DTCs, is topologically different in BPI and BPII. In particular, the disclination lines in BPI do not touch, while in BPII they intersect in four-way junctions and form two independent sets of interpenetrating defect networks, similar to the two opposite labyrinths of passage in gyroid phases. Hence, it is not possible to preserve the topology of the disclination network while transitioning from BPI to BPII. But a martensitic transformation of the DTCs through small local rearrangements seems to imply exactly the opposite, i.e. that there is a certain degree of topological similarity between BPI and BPII, as well as BPIII.

Response: The authors would like to thank the reviewer for the helpful comment.

Comment 4: However, Xiao Li et al. reported that at the BPII-BPI transition a spontaneous and sudden reconfiguration takes place where the **disclination lines** disappear, reappear, fuse and merge to produce a new crystalline structure. A possible explanation for this striking behavior, which was found in simulations and is also consistent with Li's experimental observations, could be that around disclination lines the liquid crystal can be thought of as existing in its high temperature phase, therefore offering virtually no resistance to reshaping and morphing. In contrast, the DTCs are the energetically most stable conformations. It is therefore plausible that the DTC structure remains largely intact while it undergoes a collective reorganization over larger sample areas. This could happen in a manner of recruiting new molecules to become part of DTCs and expelling others from DTCs, simply through a change in their average orientation with respect to their neighbours and akin to a travelling wave that transports only energy, but displaces no mass.

Response: The authors would like to thank the reviewer for the helpful comment. The rearrangement behaviors of DTCs are presented in the following.

The structures of the blue phase I (BPI) with body-centered (BCC) unit cells, blue phase II (BPII) with simple cubic (SC) unit cells, and blue phase III (BPIII) with spaghetti-like tangles spontaneously consist of double-twist cylinders (DTCs) and a disclination network (Figure 1). In Figs. 1A₁-A₂, the blue cylinders are the DTCs of BPII while the red rods represent the disclination lines that intersect in four-way junctions. In Figs. 1B₁-B₂, the green cylinders and the red rods are the DTCs and disclination lines of BPI, respectively, forming two independent sets of interpenetrating defect

networks.

Figure 1 DTCs and the corresponding disclination lines formed in the cubic structures as unit cells of BPII and BPI. (A₁-A₂) BPII and (B₁-B₂) BPI unit cells viewed along different directions.

The liquid crystal (LC) molecules in BPLCs form DTCs in which the director is parallel to the cylinder axis at the center, and twists by 45° along all directions perpendicular to it¹. The molecules rotate to form a helical structure, which further assembly in two orthogonal directions forming DTCs. In BPLCs, molecule diffusion occurs constantly and flows in three-dimensional (3D) crystalline. The preferred orientation of LC molecules in BPLCs depends on their position; that is, once the positions of the molecules in the DTCs are determined, their average orientation in the DTCs is also found.

These DTCs arise from a balance of long-range elastic distortions and short-range enthalpic contributions to the free energy, adopting a cubic lattice, which is interspersed by an ordered network of topological defects (disclination lines).²

In the space between DTCs, molecular orientations cannot extend smoothly, and DTCs cannot fill the space evenly. Dubbed geometrical or topological “frustration” of DTCs lead to a cubic lattice of defect lines called disclination lines. These regions of molecular misalignment are energetically most disfavored in the blue phases³. Herein, the disclination lines are topological defects (Fig. 2A), which are distinct from the structural defect⁴. The disclination lines in BPLCs have a typical diameter of approximately ten nanometers⁵, which can be further used to effectively template a variety of dynamic micro- and nanoparticle 3D assemblies⁶. In the BPI, the disclination lines are straight (Fig. 2B) and do not intersect with each other, while the disclination lines of BPPII form a four-arm junction at the center of the unit cell (Fig. 2C)². The disclination lines for BPPIII form amorphous networks with a symmetry similar to that of the isotropic state (Fig. 2D).

Figure 2 (A) DTCs with axes perpendicular to the plane of the paper present topological defects, where the DTCs cannot fill the space evenly. Simulated results of the disclination network for (B) BPI, (C) BPII, and (D) BPIII. Reproduced based on Ref. ⁵.

In this work, the phase transformation of BPI/BPII is confirmed to be a thermoelastic martensitic transformation (see pages 18-19, lines 366-387 in the revised manuscript). The phase transformations of BPI/BPIII and BPII/BPIII are also confirmed as DLPT (see page 19, lines 394-402 in the revised manuscript) for the diffusionless behavior of DTCs (Figs. S3-7, S43 in the supporting information). Herein, the rearrangement behaviors of DTCs are discussed.

i) **The thermoelastic martensitic phase transformation of BPII/BPI**

First, the phase transformation of BPI/BPII was proven to be a thermoelastic martensitic transformation. To demonstrate the rearrangement of DTCs during the transformation of BPII/BPI, the projected models presented in Figures 3-5 are based on the experimental results of the lattice

constants corresponding to BPI and BPII, which are measured by syn-SAXS, resulting in 254.48 nm of BPI and 167.3 nm of BPII.

Figure 3 Lattice orientation relationships between BPI and BPII experiencing a thermoelastic martensitic transformation. (A₁) BPII unit cells with the {100} orientation out-of-plane (BPII_{100}). The black solid lines connect the center of DTCs with the vertical orientation in BPII_{100}. (A₂) BPI unit cells with the {110} orientation out-of-plane (BPI_{110}). The black dash lines connect the center of DTCs with the tilt orientation in BPI_{110}. (A₃) Schematic of the angle between the vertical DTCs of BPII and tilt DTCs of BPI, which is calculated to be 9.74°. (B₁) {100} crystal plane of BPII ({100}_{BPII}) is approximately parallel to the {110} crystal plane of BPI ({110}_{BPI}) while there is a small angle of 9.74° between them. (B₂) Schematic illustration of the orientation relationship between BPI and BPII where {110}_{BPII} is parallel to {211}_{BPI}. (C₁) Angle-resolved microspectroscopy (ARM) result of BPLCs captured at 77.0 °C (Stage III) exhibit well-matched dips of the {110}_{BPI} streak with the {100}_{BPII} streak. The arrows highlight the centers of the streaks. (C₂) ARM result of BPLCs captured at 74.8 °C (Stage IV) exhibits a slightly mismatched center of the {110}_{BPI} streak in relative to {100}_{BPII}. Because the center of BPI does not shift, the mismatch may be caused by the rotation of {100}_{BPII}, resulting in a small angle between the approximately paralleled {100}_{BPII} and {110}_{BPI}. The arrows highlight the centers of the streaks. (D₁) Syn-SAXS pattern measured from the BPLCs with BPI_{110} and BPII_{100} in coexistence, which is polymer-stabilized at 77.0 °C. (D₃) The speckles

of $\text{BPI}_{\{211\}}$ and $\text{BPII}_{\{110\}}$ are relatively close but do not overlap at 77.0 °C, indicating that the $\{110\}_{\text{BPII}}$ is approximately parallel to $\{211\}_{\text{BPI}}$ with a small angle. (D₂) Syn-SAXS pattern measured from the BPLCs with $\text{BPI}_{\{110\}}$ and $\text{BPII}_{\{100\}}$ in coexistence, which is polymer-stabilized at 75.8 °C. (D₄) The speckles of $\{211\}_{\text{BPI}}$ and $\{110\}_{\text{BPII}}$ are overlapped, indicating that $\{110\}_{\text{BPII}}$ and $\{211\}_{\text{BPI}}$ are parallel (Fig. S52A₁ in the supporting information).

As shown in Figures 3A₁-A₂, the black solid lines connect the centers of the DTCs with a vertical orientation in $\text{BPII}_{\{100\}}$ while the black dashed lines connect the centers of the DTCs with tilt orientation in $\text{BPI}_{\{110\}}$. An angle of 9.74 ° ($=\arccos\frac{\sqrt{3}}{3}-45^\circ$) (Fig. 3A₂) between the black solid lines highlighted in $\text{BPII}_{\{100\}}$ (Fig. 3A₁) and the black dashed lines in $\text{BPI}_{\{110\}}$ (Fig. 3A₂) are observed in the projected models. Thus, a rotation of 9.74 ° (Fig. 3A₃) is required during the thermoelastic martensitic transformation between $\text{BPI}_{\{110\}}$ and $\text{BPII}_{\{100\}}$. It is hypothesized that the vertical DTCs in $\text{BPII}_{\{100\}}$ tend to form tilt DTCs in $\text{BPI}_{\{110\}}$ during the cooling process. The diffusionless fuse and merge of DTCs rearrange through the local molecular twist and reorientation. The rotation of the azimuthal angle was further proved by syn-SAXS and angle-resolved microspectroscopy (ARM) results.

The $\text{BPI}_{\{110\}}$ and $\text{BPII}_{\{100\}}$ streak dips are observed to be well-matched in the early stage [77.0 °C in (Fig. 3C₁)] of the phase transformation, proving $\{110\}_{\text{BPI}} // \{100\}_{\text{BPII}}$. When the phase transition is in the last stage [(74.8 °C in (Fig. 3C₂))], a small deviation of the dips is observed (Fig. 3 C₁), resulting in a small angle appearing between $\{110\}_{\text{BPI}}$ and $\{100\}_{\text{BPII}}$. In this case, $\{211\}_{\text{BPI}} // \{110\}_{\text{BPII}}$ was achieved (Figs. 3B₁-B₂). The dip of $\text{BPI}_{\{110\}}$ does not change while that of $\text{BPII}_{\{100\}}$ is rotated, which indicates that the rotation of the parent lattices is required during the thermoelastic martensitic transformation.

Simultaneously, the rotation of the parent BPII phase and the resulting $\{211\}_{\text{BPI}} // \{110\}_{\text{BPII}}$ can be further confirmed by the syn-SAXS results. In the early stage of phase transition, an approximately parallel relationship between $\{211\}_{\text{BPI}}$ and $\{110\}_{\text{BPI}}$ can be observed, corroborated by the nearly but not overlapping speckles, as shown in Figs. 3D₁ and the inserted D₃. However, $\text{BPII}_{\{100\}}$ rotates during the phase transformation, resulting in $\{110\}_{\text{BPII}}$ being parallel to $\{211\}_{\text{BPI}}$ (Figs. 3D₂ and inserted D₄), which is confirmed by the overlap of the speckles of $\{211\}_{\text{BPI}}$ and $\{110\}_{\text{BPII}}$ in a monodomain PS-BPLCs hybrid with $\text{BPI}_{\{110\}}$ and $\text{BPII}_{\{100\}}$ phases.

Figure 4 (A₁) BPII unit cells with the {100} orientation out-of-plane. The axis of horizontal DTCs is highlighted by black solid lines. (A₂) BPI unit cells with the {110} orientation out-of-plane. The axis of horizontal DTCs is highlighted by black dashed lines. (A₃) Horizontally oriented DTCs of BPII_{100} are parallel to that of BPI_{110}. The interval of horizontal DTCs between BPI_{110} and BPII_{100} are similar. (B₁) TEM image of the interface between BPI_{110} and BPII_{100} (Fig. S26B in the supporting information). (B₂) Magnified TEM image of the interface between BPI_{110} and BPII_{100} (Fig. S32C₁ in the supporting information). (B₃) Theoretically predicted arrangement of the DTCs near the interface (Fig. S32C₄ in the supporting information).

As shown in Figure 4, nearly one-third of the DTCs in BPI_{110} (horizontal DTCs of BPI_{110} in Fig. 4A₂) transfer from the horizontal DTCs of BPII_{100} (Fig. 4A₁). Because the spaces between adjacent horizontal DTCs in BPII_{100} are similar to those of BPI_{110} (Fig. 4A₃), the formation of a coherent lattice at the interface between BPI_{110} and BPII_{100} is predicted. Figs. 4B₁-B₃ present the microstructures measured by TEM and the theoretically predicted arrangement of DTCs near the interface between BPII_{100} and BPI_{110}. A clear boundary and coherent lattice are observed in the TEM image (Figs. S26B, S32C, and S43 in the supporting information). Furthermore, the 2-θ values measured by syn-SAXS corresponding to {110}_{BPII} and {211}_{BPI} are similar (inserted Fig. 3D₃) (Figs. S44C₃, E₃, S45C₁-D₁, S46, S52D₁-D₄ in the supporting information), confirming the coherence of the crystal lattice between {110}_{BPII} and {211}_{BPI}, providing evidence for the thermoelastic martensitic transformation.

Figure 5 (A) BPII unit cells with the $\{100\}$ orientation out-of-plane. The solid black circles highlight the position of DTCs with vertical orientation. (B) BPI unit cells with the $\{110\}$ orientation out-of-plane. The black dashed ellipses highlight the position of DTCs with tilt orientation. (C) The relationship of the vertical DTCs between $BPI_{\{110\}}$ and $BPII_{\{100\}}$ is evaluated by the quantity and position of DTCs. The orange arrows highlight the predicted tendency of the rearrangement of DTCs during the phase transformation from BPII to BPI during the cooling process.

As shown in the projected models in Figure 5, the vertical DTCs (solid circles) in $BPII_{\{100\}}$ (Fig. 5A) tend to transfer to the tilted DTCs (dashed circles) in $BPI_{\{110\}}$ (Fig. 5B). The orange arrows in Fig. 5C indicate the probable tendency of the rearrangement of DTCs. However, the total number of tilt DTCs in $BPI_{\{110\}}$ is greater than the vertical DTCs of $BPII_{\{100\}}$ and the extra yellow ellipses shown in Fig. 5C may transfer from the adjacent horizontal (longitudinal) DTCs of $BPII_{\{100\}}$.

Figure 6 Models highlight the proposed mechanism of the collective move of DTCs during the thermoelastic martensitic transformation from $BPII_{\{100\}}$ with 16 unit cells of $(2 \times 2 \times 4)$ to $BPI_{\{110\}}$. (A) Pure $BPII_{\{100\}}$ with 16 unit cells. (B-E) The process of thermoelastic martensitic transformation from $BPII_{\{100\}}$ to $BPI_{\{110\}}$. (F) Pure $BPI_{\{110\}}$.

Based on the experimental results and analysis shown in Figures 3-5, the entire DTC

rearrangement processes is presented in Figure 6 during the thermoelastic martensitic transformation from $\text{BP}_{\text{II}\{100\}}$ (with 16 unit cells of $(2 \times 2 \times 4)$) to $\text{BP}_{\text{I}\{110\}}$. The following DTC behaviors were confirmed owing to their rearrangement during the thermoelastic martensitic transformation of $\text{BP}_{\text{II}}/\text{BP}_{\text{I}}$: (i) the parent lattice of BP_{II} needs to rotate 9.74° ; (ii) nearly one-third of the $\text{BP}_{\text{II}\{100\}}$ DTCs (the horizontal DTCs) tend to transfer to the horizontal DTCs of $\text{BP}_{\text{I}\{110\}}$, resulting in a coherent lattice near the interface of $\text{BP}_{\text{II}}/\text{BP}_{\text{I}}$, and (iii) the vertical DTCs in $\text{BP}_{\text{II}\{100\}}$ tend to transfer to the tilt DTCs in $\text{BP}_{\text{I}\{110\}}$. The total number of tilt DTCs in $\text{BP}_{\text{I}\{110\}}$ is greater than the vertical DTCs of $\text{BP}_{\text{II}\{100\}}$. The extra tilt DTCs of $\text{BP}_{\text{I}\{110\}}$ may be transferred from the adjacent horizontal (longitudinal) DTCs of $\text{BP}_{\text{II}\{100\}}$.

ii) The DLPT of $\text{BP}_{\text{III}}/\text{BP}_{\text{I}}$ and $\text{BP}_{\text{III}}/\text{BP}_{\text{II}}$

Furthermore, the phase transformations of $\text{BP}_{\text{III}}/\text{BP}_{\text{I}}$ and $\text{BP}_{\text{III}}/\text{BP}_{\text{II}}$ are also confirmed as DLPT because the DTCs of BP_{III} do not diffuse when transferred to BP_{II} or BP_{I} (Figs. S3-7 and S43 in the revised manuscript). It was found that a thin shell of BP_{III} always exists between the isotropic state and BP_{II} or the isotropic state and BP_{I} , forming $\text{BP}_{\text{III}}/\text{BP}_{\text{II}}$, $\text{BP}_{\text{III}}/\text{BP}_{\text{I}}$, and $\text{BP}_{\text{III}}/\text{BP}_{\text{II}}/\text{BP}_{\text{I}}$ core-shell configurations. BP_{II} or BP_{I} nuclei cannot directly form in the isotropic state (Figs. 1D₂, S6F, S11F-G in the revised manuscript and supporting information), suggesting that BP_{II} and BP_{I} cannot transfer directly from the isotropic state without DTCs; that is, the formation of BP_{II} and BP_{I} with DTCs arranged in cubic symmetry benefit from the pre-formed DTCs of BP_{III} . In addition, the positions of DTCs were investigated by TEM at the interfaces of $\text{BP}_{\text{III}}/\text{BP}_{\text{II}}$, and $\text{BP}_{\text{III}}/\text{BP}_{\text{I}}$ (Fig. S43 in the revised manuscript). Owing to the apparent interfaces observed by the TEM between BP_{III} and BP_{II} (Figs. S29, S32E, and S43B in the supporting information), the grid with four-fold symmetry highlighted by the dotted lines belong to $\text{BP}_{\text{II}\{100\}}$. The curved lines on the side of the grid highlight the DTCs crossing the interface and simultaneously exist in both $\text{BP}_{\text{II}\{100\}}$ and BP_{III} . It that the diffusionless reconfiguration of DTCs is confirmed to occur during the phase transformation between BP_{II} and BP_{III} rather than as a process of diffusion first and then reorganization. Considering the phase transformation between BP_{III} and BP_{I} (Fig. S43C in the supporting information), a similar phenomenon was observed for the interface of $\text{BP}_{\text{II}}/\text{BP}_{\text{III}}$. Therefore, the phase transformation between BP_{III} and BP_{I} is also diffusionless.

In summary, based on the core-shell configurations of $\text{BP}_{\text{III}}/\text{BP}_{\text{II}}$, $\text{BP}_{\text{III}}/\text{BP}_{\text{I}}$, and $\text{BP}_{\text{III}}/\text{BP}_{\text{II}}/\text{BP}_{\text{I}}$ observed by POM, and the arrangement of DTCs directly observed by TEM in real-

space, the phase transformations of BPIII→BPII and BPIII→BPI are proven as DLPT.

Comment 5: My main concern is that the experimental system is not sufficiently distinctly described as different from a pure blue phase. It consists of almost two thirds (65% wt) of polymer and cross-linker. From a practical point of view, this is of course necessary for subjecting the same samples to further experimental analysing techniques that are required. It is, however, really unclear how the dominating presence of polymer affects the phase transition behaviour. I find it problematic to apply the results directly to support statements about the general phase transition behaviour of pure blue phases, even though the individual phases are identified as BPs and the results seem to agree well with the previous results of Xiao Li et al., who studied pure, mono-crystalline blue phases that were grown using chemically patterned surfaces. I feel that this feature may be also responsible for some of the observed peculiarities like the coherence of phases across the interface separating two blue phases. If my back-of-the-envelope calculation is correct, the latent heat involved in these coherent structures is on the order of 10^6 kB T or even more and therefore very large. The authors should please emphasize why they think that despite the different experimental system these results also apply to pure BPs.

Response: The authors would like to thank the reviewer for this critical comment. There is a misunderstanding caused by the unclear demonstration of the authors. In this work, the major findings of the DLPT were studied in pure BPLCs with a liquid-like nature (before polymer-stabilization, termed BPLCs in the revised manuscript) (Figs. 1A₂-E₂, S1-9, S11, S13-15, S50-51 in the revised manuscript and supporting information) rather than the polymer-stabilized BPLCs (after polymer-stabilization, termed PS-BPLC in the revised manuscript) that coexisted with the polymer content.

To the best of our knowledge, typical pure BPLCs should meet the following conditions:

- (i) They are a special class of chiral nematics that mostly appear between the isotropic state and cholesteric LC, which are highly ordered at the two levels of molecules and DTCs⁷.
- (ii) Molecules with rod-like shapes have an orientational order at the nanometer scale, and the molecules rotate to form helical vortices³.
- (iii) In BPLCs, chirality leads to helical ordering in two orthogonal directions, which can be visualized as DTCs^{2,6}.
- (iv) The DTC structures form periodic 3D orientational textures with cubic symmetry³; the

DTCs are spontaneously stacked to fill a three-dimensional space, forming a simple cubic (SC) lattice for BPII with a space group of O^2 (P4₂32), a body-centered cubic (BCC) lattice for BPI with a space group of O^8 (I4₁32), or a spaghetti-like tangle of BPIII (also termed fog phase) that has the same symmetry as the isotropic state⁶.

- (v) In this case, no polymer content should be included in the hybrid LC system, and the molecule diffusion occurs constantly; and their preferred orientation depends on their position.
- (vi) The characteristic periods of BPI and BPII are several hundred nanometers in size, allowing for light manipulation at visible, infrared, or ultraviolet wavelengths.
- (vii) Red, green, and blue platelet textures are observed by POM with crossed polarizers.

In this work, the BPLCs prior to polymerization meet all aforementioned conditions and are thus considered as pure BPLCs. The DLPT conclusion based on the BPLCs before polymerization in this work can be adapted to the pure BPLCs system. The hybrid LCs used to self-assemble the BPLCs were HTG135200 (30 wt%), C6M (61 wt%), R5011(3.5 wt%), TMPTA (4.0 wt%), and I-651 (1.5 wt%), which contain the reactive monomer of C6M and an initiator of I-651 but no polymer in the hybrid LC system before polymerization.

Figure 7 Optic textures varied with temperature when cooled from 78.5 °C (Stage I) to 74.0 °C (Stage V) at a rate of 0.05 °C/min. (A₁-A₅) Pure BPLCs exhibit DLPT properties. (B₁-B₅) PS-BPLCs photopolymerized at stage IV without changes with temperature.

In addition, the mixed LC components can be photopolymerized to stabilize the microstructures of pure BPLCs, resulting in polymer-stabilized BPLCs (PS-BPLCs). The main function of the PS-BPLCs presented in this work is to thoroughly investigate the phase transition behavior of BPLCs in submicrometer real- or reciprocal-space. Besides, once the BPLCs are photopolymerized, no changes including the DLPT occur in the PS-BPLCs (Figs. 7B₁-B₅) compared to the pure BPLCs (Figs. 7A₁-A₅). Furthermore, PS-BPLCs can stabilize in an ultra-wide temperature range of greater than 500 °C

(from -190 °C to 340 °C) (Figs. S17–22 in the supporting information), which can preserve the microstructures and optical properties of pure BPLCs (Figs. S12, S16B, S17-30, S32, S35-40, S42-48, S60-61, S63-66, S69-70 in the supporting information).

After polymerization, the pure BPLCs were stabilized as PS-BPLCs, and the helical vortices of molecules, arrangement of DTCs, typical textures and pseudogaps of pure BPLCs were frozen to further characterizations. The helical-vortex oriented molecules in the DTC microstructures are proved by rotating the polarizer of POM (Figs. S3 and S6 in the supporting information). The twisted molecules in the DTCs can be conveniently distinguished from the isotropic state owing to their optical activity, which renders different colors (domains exhibit yellowish-brown and dark blue)⁸⁻¹¹. In addition, the DTC structures were directly observed by TEM (Figs. S23-43 in the supporting information) which sufficiently agree well with the theoretically predicted arrangement of the DTCs (Fig. S32 in the supporting information). The periodic cubic lattices of BPI and BPII were proved by syn-SAXS (Figs. S44-48 in the supporting information) and ARM (Figs. S49-67 in the supporting information).

In this work, the fundamental properties of the DLPT were observed in pure BPLCs (before polymerization) with a liquid-like nature (Figs. 1A₂-E₂, S1-9, S11, S13-16, S50-51 in the revised manuscript and supporting information). The PS-BPLCs (after polymerization) without phase transformation are adopted to freeze the microstructures at certain stages of pure BPLCs for thorough characterization and application. We have added it in the revised manuscript. **Please see:** Page 7, lines 159-162.

Comment 6: To call the transition diffusionless triggers simply the wrong associations, but follows the terminology of martensitic transformations and Xiao Li et al. As in any liquid matter, there is of course diffusion present at the molecular level. But the claim is rather that the emerging, mesoscopic, double-twist cylinder (DTC) structures do not shape and reform as a whole through diffusion, but through large and collective reorganization. This distinction, and the presence of diffusion at the molecular level, should be made as clear as possible to avoid confusion and unnecessary criticism.

Response: The authors would like to thank the reviewer for the helpful comment. The manuscript is unclear in distinguishing the diffusion at the molecular level and the collective reorganization of DTCs with tens of nanometers. We have carefully revised the manuscript according to the comments

of the reviewer.

In this work, molecule diffusion occurs constantly and flows in a three-dimensional (3D) crystalline. The preferred orientation of LC molecules in BPLCs depends on their position; that is, once the positions of the molecules in the DTCs are determined, their average orientation in the DTCs is also determined. The formed DTCs are found to reconfigure collectively and are highly coordinated during the phase transition process. The relative demonstrations are revised in the manuscript.

BPLCs are analogous to atomic crystals which are highly ordered structures at both the molecular (orientational molecular order) and mesoscopic scales (DTCs). The term “diffusionless” is widely used in metal, alloys, or ceramic materials, and a classic diffusionless transformation in atomic crystals is martensitic. Herein, “diffusionless” indicates that the DTCs, regarded as the building blocks, exhibit a diffusionless and collective transformation during the phase transition process among BPIII, BPII, and BPI, which are analogous to the atoms during diffusionless transformation of an atomic crystal. In this work, not only the thermoelastic martensitic transformation of BPII→BPI but also the transformation of BPIII→BPI and BPIII→BPII were regarded as DLPT. Thus, the terminology of “diffusionless” (including of the phase transformation of BPIII/BPI, BPIII/BPII, and BPII/BPI) was adopted rather than the “martensitic transformations” (only for phase transformation of BPII/BPI). We have added the relative content in the revised manuscript.

Furthermore, although the previous work has investigated the simulation of martensitic transformation which lacks the experimentally dynamic track of the DTC arrangement in real- or reciprocal-space. Herein, several conclusions are made in our work which were not presented in the previous paper:

- (i) DLPT of BPIII-to-BPI and BPIII-to-BPII and the thermoelastic properties of the martensitic transformation;
- (ii) Thermoelastic properties of BPII-to-BPI are experimentally observed by a coherent lattice, twins, completely reversible phase transformation, and surface relief.
- (iii) Applications based on DLPT including ternary QR codes (Fig. 4 in revised manuscript), micro-area lasing (Fig. 5 in revised manuscript), and fabrication of BPLCs with large domain sizes (Figs. S14-15 in the supporting information);
- (iv) Three types of core-shell configurations of BPIII/BPII/BPI, BPIII/BPII, and BPIII/BPI, which are confirmed by TEM, POM, and ARM, provide opportunities to investigate the

- interface among BPLCs;
- (v) Phase transition from BPIII directly to BPI;
 - (vi) Stability of PS-BPLCs in the temperature range of (-190 to 340 °C) is achieved to freeze the microstructures for investigation by TEM and syn-SAXS;
 - (vii) Two-step successive DLPT consisting of BPIII-to-BPII and BPII-to-BPI as well as three-step successive DLPT consisting of BPIII-to-BPI, BPIII-to-BPII, and BPII-to-BPI;
 - (viii) The distribution and rearrangement of DTCs during the DLPT among BPIII, BPII, and BPI are tracked by TEM in real-space and syn-SAXS in reciprocal-space;
 - (ix) Pure BPLCs exhibit five-stages of phase transitions which are classified based on the coexistence of BPI, BPII, and BPIII (Fig. S3 in supporting information).
 - (x) Relationship between crystalline planes and several behaviors of the rearrangement of DTCs.

Besides, a thorough comparison between the manuscript and the experiment by Xiao Li et al., PNAS 2017 is presented in Table 1.

Table 1. A through comparison between this work and the experiments by Xiao Li et al., PNAS 2017

No.		This work	Xiao Li et al., PNAS 2017
1	Fabricat.	Both poly- (> 200 μm) and single-domain (3×2 cm) BPLCs with large domain size are fabricated. The single-domain BPLCs are constructed in LC cells with two substrates covered by rubbed polyimide layers.	Single-domain BPLCs are prepared using a chemically patterned substrate underneath the hybrid liquid crystals. The size of the single-domain BPLCs is as large as several millimeters.
2	Charact.	The properties of DLPT are not only investigated based on pure BPLCs with liquid nature but also in free-standing PS-BPLCs, providing opportunities for further characterization of micro-structures including POM, TEM, syn-SAXS, and ARM.	Pure BPLCs with liquid-like nature were obtained on solid surfaces in LC cells. POM characterization was performed. No application or other characterization was mentioned.
3		The distribution and rearrangement of DTCs during the diffusionless phase transition processes among BPI, BPII, and BPIII were directly tracked by TEM in real-space and syn-SAXS in reciprocal-space.	Only the POM characterization was carried out on the phase transformation of BPI/BPII.
4	Results	The pure BPLCs exhibit five-stages phase transitions classified based on the coexistence of BPI, BPII, and BPIII: Stage I: BPIII domains. Stage II: BPIII/BPII core-shell configurations. Stage III: BPIII/BPII, BPIII/BPI, and BPIII/BPII/BPI core-shell configurations. Stage IV: BPIII/BPII/BPI core-shell configuration and BPIII domain. Stage V: BPI domains with distinct orientation.	Not mentioned.
5		Several behaviors of the rearrangement of DTCs are confirmed during the thermoelastic martensitic transformation of BPII/BPI: A. The parent lattice of $\text{BPII}_{\{100\}}$ needs to rotate 9.74° to transfer to $\text{BPI}_{\{110\}}$. B. Nearly one-third of the DTCs of $\text{BPII}_{\{100\}}$ (horizontal DTCs) tend to transfer to the horizontal DTCs of $\text{BPI}_{\{110\}}$, resulting in the coherent lattice near the interface of BPII/BPI. C. The vertical DTCs in $\text{BPII}_{\{100\}}$ tend to transfer to the tilt DTCs in $\text{BPI}_{\{110\}}$. The total number of tilt DTCs in $\text{BPI}_{\{110\}}$ is more than the vertical DTCs of $\text{BPII}_{\{100\}}$. The extra tilt DTCs of $\text{BPI}_{\{110\}}$ may transfer from the horizontal DTCs of $\text{BPII}_{\{100\}}$.	Not mentioned.
6		The phase transformation during BPII-to-BPI is further proved as a thermoelastic martensitic transformation based on experimental observation of coherent lattice, twins, completely reversible phase transformation and surface reliefs.	The martensitic transformation is predicted by the simulation based on reconfiguration of topological defect network during BPII-to-BPI phase transformation.

No.	This work	Xiao Li et al., PNAS 2017
7	Three types of core-shell configurations are observed during the phase transition process: BPIII/BPI, BPIII/BPII, and BPIII/BPII/BPI core-shell configurations, providing the grain boundaries for direct observation of POM, ARM, and TEM. The coexistence of BPIII, BPII, and BPI provides the opportunity to achieve binary and ternary temperature-switchable quick response (QR) codes.	In single-domain BPLCs, the hybrid phase consists of BPI and BPII can not be distinguished to study the phase transition behavior. The grain boundary between BPI and BPII cannot be observed experimentally by POM. Thus, the study of initial crystallization kinetics of BPLCs including the nucleation and growth is not men-
8	Results Two types of successive DLPT are confirmed in our experiment when the BPLCs move from Stage I to V (Fig. S11 in the supporting information): (i) The two-step process consists of BPIII→BPII and BPII→BPI sequentially, resulting in sequential transitions from BPIII to BPIII/BPII and BPIII/BPII/BPI core-shell configurations and finally to BPI. (ii) The three-step process consists of BPIII→BPI, BPIII→BPII, and BPII→BPI sequentially, which causes sequential transitions from the BPIII domain to BPIII/BPI and BPIII/BPII/BPI core-shell configurations and finally to BPI .	Not mentioned.
9	DLPT among BPI/BPIII, BPII/BPIII, and thermoelastic martensitic transformation of BPI/BPII are investigated both in poly- and single-domain BPLCs.	The thermoelastic properties of martensitic transformation of BPI/BPII and DLPT properties of BPI/BPIII or BPII/BPIII are not mentioned.
10	A series of applications had been developed such as temperature-switchable binary and ternary QR codes (Fig. 4 in the manuscript), micro-area lasing (Fig. 5 in the manuscript), and fabrication of BPLCs with large domain sizes (Figs. S14-15 in the supporting information).	Not mentioned.
11	Applica. This work has provided a possible route for basic understanding of the crystalline transformations. Owing to the high-resolution of spectra and evolution of lasing performance in BPI, it is found for the first time that the thermoelastic martensitic transformation appears to be an unprecedented dual-stage process, which is generally difficult to characterize in metal or ceramic materials owing to their complex characterization conditions. During this process, the fresh-formed crystalline grains exhibit a high degree of order, and the quality of the crystal lattice can be further improved by the growth of martensitic grains (Fig. 5 in the revised manuscript)	Not mentioned.

Please see: “In this work, we demonstrate the structural units of the BPLC, i.e., the double-twisted cylinders

(DTCs) of molecular orientation, exhibit diffusionless behavior with collective and highly coordinated motion during the phase transformation of BPLCs.” (Page 2, lines 38-40). “The DTCs, which are analogous to the atoms of the atomic crystals...” (Page 3, lines 67-68). “This study reveals the nucleation mechanism and rearrangement behaviors of DTCs during successive DLPTs...” (Page 4, lines 86-87). “Herein, DTCs serve as building blocks that are analogous to atoms for atomic crystals” (Page 5, lines 115-117). “In this case, molecules can diffuse constantly and flows in a three-dimensional (3D) crystalline and their directors determined by their...through diffusion” (Page 10, lines 218-221).

Comment 7: A minor point: On p.7 the authors say the transition between BPI and BPII is reversible without hysteresis. As a weak first-order transition, which the pure BPI-BPII transition undoubtedly is, I think there has to be hysteresis. Moreover, the authors say in Fig. S8 that they flow cold air over the sample to trigger the transition. This sounds like a rather uncontrolled modulation of temperature, which may obscure the hysteretic behaviour. It could also be that this is somehow caused by the high level of polymer present. I think this part should be accordingly reworded.

Response: The authors would like to thank the reviewer for the critical comment. In this work, the BPLCs used to measure the properties of hysteresis are pure BPLCs before photo-polymerization, which were not stabilized by photo-polymerization and there was no polymer.

In addition, the phase transition between BPI and BPII is proven to be a thermoelastic martensitic transformation. The phase transformation behavior of BPI/BPII is completely reversible (Figs. 4, S8-9 in the revised manuscript and supporting information) and can be considered as a weak first order transformation¹² without apparent hysteresis.

Figure 8 Completely reversible thermoelastic martensitic transformation between BPII and BPI in BPIII/BPII/BPI core-shell configuration (Stage IV) with distinct temperature change rates of (A) 0.1, (B) 0.3, (C) 0.5, (D) 1.0, and (E) 2.0 °C/min

Except for the flow of cold air over the sample (Movie 2 and Fig. S8 in the supporting information), several detailed heating and cooling cycles were performed on the pure BPLCs at the rates of 0.1, 0.3, 0.5, 1.0 and 2.0 °C/min to clearly investigate the hysteresis properties (Fig. 8). Temperature was controlled by the Linkam heating stage equipped with a liquid nitrogen cooling system. The completely reversible phase transition between BPI and BPII observed by POM without apparent temperature hysteresis provides significant evidence of thermoelastic martensitic

transformation. We have added the relative data in the supporting information. Please see Figure S9 in the revised manuscript.

- 1 Meiboom, S., Sethna, J. P., Anderson, P. W. & Brinkman, W. F. Theory of the Blue Phase of Cholesteric Liquid Crystals. *Physical Review Letters* **46**, 1216-1219, doi:10.1103/PhysRevLett.46.1216 (1981).
- 2 Li, X. *et al.* Mesoscale martensitic transformation in single crystals of topological defects. *Proc. Natl. Acad. Sci. U S A* **114**, 10011-10016, doi:10.1073/pnas.1711207114 (2017).
- 3 Ravnik, M., Alexander, G. P., Yeomans, J. M. & Zumer, S. Three-dimensional colloidal crystals in liquid crystalline blue phases. *Proc Natl Acad Sci U S A* **108**, 5188-5192, doi:10.1073/pnas.1015831108 (2011).
- 4 Zapotocky, M., Ramos, L., Poulin, P., Lubensky, T. C. & Weitz, D. A. Particle-stabilized defect gel in cholesteric liquid crystals. *Science* **283**, 209-212, doi:10.1126/science.283.5399.209 (1999).
- 5 Henrich, O. & Marenduzzo, D. The secret of the blue fog. *Phys. World* **30**, 25-39 (2017).
- 6 Gharbi, M. A. *et al.* Reversible Nanoparticle Cubic Lattices in Blue Phase Liquid Crystals. *ACS Nano* **10**, 3410-3415, doi:10.1021/acs.nano.5b07379 (2016).
- 7 Chen, C. W. *et al.* Large three-dimensional photonic crystals based on monocrystalline liquid crystal blue phases. *Nat. Commun.* **8**, 727-735, doi:10.1038/s41467-017-00822-y (2017).
- 8 Gandhi, S. S. & Chien, L. C. Unraveling the Mystery of the Blue Fog: Structure, Properties, and Applications of Amorphous Blue Phase III. *Adv. Mater.* **29**, 1704296-1704309, doi:10.1002/adma.201704296 (2017).
- 9 Chiang, I. H. *et al.* Broad ranges and fast responses of single-component blue-phase liquid crystals containing banana-shaped 1,3,4-oxadiazole cores. *ACS Appl Mater Interfaces* **6**, 228-235, doi:10.1021/am403976a (2014).
- 10 Kim, M. S. & Chien, L. C. Topology-mediated electro-optical behaviour of a wide-temperature liquid crystalline amorphous blue phase. *Soft Matter* **11**, 8013-8018, doi:10.1039/c5sm01918d (2015).
- 11 Dressel, C. *et al.* Helical Networks of π -Conjugated Rods – A Robust Design Concept for Bicontinuous Cubic Liquid Crystalline Phases with Achiral Ia $\bar{3}$ d and Chiral I23 Lattice. *Advanced Functional Materials* **30**, doi:10.1002/adfm.202004353 (2020).
- 12 Chen, C.-W. *et al.* Temperature dependence of refractive index in blue phase liquid crystals. *Optical Materials Express* **3**, doi:10.1364/ome.3.000527 (2013).

Reviewers' Comments:

Reviewer #3:

Remarks to the Author:

The authors have addressed all my points but one, which I think is subtle, but important. I understood that the authors found the diffusionless phase transition in the liquid-like blue phase (BP) before polymer-stabilization. But while I agree that the authors have studied a BP, I do not think they studied a PURE BP.

The vast majority of researchers associates with the term 'pure BP' something that consists of a very high concentration of mesogens with a few percent of chiral dopant. This is in stark contrast to the authors' system, where almost two thirds are made up of polymer.

Hence, the word 'pure' should be removed from the main manuscript (line 84) and the supplement (242, 378, 379, 382, 969 and 970). Otherwise, I have no objections to recommend this excellent piece of work for publication in Nat. Commun.

Dear referees:

Thank you very much for the comments of our manuscript titled “Diffusionless Transformation of Soft Cubic Superstructure from Amorphous, Simple Cubic to Body-Centered Cubic Phases”(manuscript No. NCOMMS-20-09527B) by Jie Liu, Wenzhe Liu, Bo Guan, Bo Wang, Lei Shi, Feng Jin, Zhigang Zheng*, Jingxia Wang*, Tomiki Ikeda and Lei Jiang. We appreciate your encouragement and the constructive evaluations. We have carefully read the comments and revised the manuscript accordingly (the revised section is highlighted in blue, and the response to the reviewer’s comments is attached below).

A list of main changes in the revised manuscript:

Page 1, the title is changed as “Diffusionless Transformation of Soft Cubic Superstructure from Amorphous to Simple Cubic and Body-Centered Cubic Phases

Page 2, Lines 28-39;

Page 3, Lines 72-73;

Page 4, Lines 74-75;

Page 18, Lines 355-356;

We would appreciate if the revised manuscript could be recommended publication in *Nature Communications*. Thank you very much for your kind consideration.

Yours sincerely,

Jingxia Wang

E-mail: jingxiawang@mail.ipc.ac.cn

Response to the reviewers:

Reviewer #3 (Remarks to the Author):

Comment 1: The authors have addressed all my points but one, which I think is subtle, but important. I understood that the authors found the diffusionless phase transition in the liquid-like blue phase (BP) before polymer-stabilization. But while I agree that the authors have studied a BP, I do not think they studied a PURE BP.

The vast majority of researchers associates with the term 'pure BP' something that consists of a very high concentration of mesogens with a few percent of chiral dopant. This is in stark contrast to the authors' system, where almost two thirds are made up of polymer.

Hence, the word 'pure' should be removed from the main manuscript (line 84) and the supplement

(242, 378, 379, 382, 969 and 970).

Response: Author would like to thank the reviewer for the value comment. We have removed the word of “pure” from the main manuscript and the supplementary information.

Comment 2: Otherwise, I have no objections to recommend this excellent piece of work for publication in Nat. Commun.

Response: Author would like to thank the reviewer for the positive comment.